# Quantifying the Empirical Wasserstein Distance to a Set of Measures: Beating the Curse of Dimensionality

**Nian Si**
Department of Management Science and Engineering
Stanford University
Huang Engineering Center, 475 Via Ortega, Stanford, California 94305, United States
`niansi@stanford.edu`

**Jose Blanchet**
Department of Management Science and Engineering
Stanford University
Huang Engineering Center, 475 Via Ortega, Stanford, California 94305, United States
`jose.blanchet@stanford.edu`

**Soumyadip Ghosh**
Mathematical Sciences
IBM Research
Yorktown Heights, NY 10598, USA
`ghoshs@us.ibm.com`

**Mark S. Squillante**
Mathematical Sciences
IBM Research
Yorktown Heights, NY 10598, USA
`mss@us.ibm.com`

## Abstract

We consider the problem of estimating the Wasserstein distance between the empirical measure and a set of probability measures whose expectations over a class of functions (hypothesis class) are constrained. If this class is sufficiently rich to characterize a particular distribution (e.g., all Lipschitz functions), then our formulation recovers the Wasserstein distance to such a distribution. We establish a strong duality result that generalizes the celebrated Kantorovich-Rubinstein duality. We also show that our formulation can be used to beat the curse of dimensionality, which is well known to affect the rates of statistical convergence of the empirical Wasserstein distance. In particular, examples of infinite-dimensional hypothesis classes are presented, informed by a complex correlation structure, for which it is shown that the empirical Wasserstein distance to such classes converges to zero at the standard parametric rate. Our formulation provides insights that help clarify why, despite the curse of dimensionality, the Wasserstein distance enjoys favorable empirical performance across a wide range of statistical applications.

## 1 Introduction

In this paper we consider the problem of projecting the empirical measure, under the Wasserstein distance, to a set of probability measures that are constrained to satisfy a family of expectations over a class of functions. We call this class of functions the "hypothesis class", examples of which include moment constraints or expectations of functions other than polynomials.

The Wasserstein distance has generated a great deal of attention in recent years across a broad spectrum of areas, ranging from artificial intelligence, learning and statistics to areas such as image analysis, economics and operations research [1, 18, 9, 12, 15]. However, despite its versatility and

modelling power, classical results on the rates of statistical convergence of the Wasserstein distance metric show that these rates scale poorly as a function of the dimension of the space [8]. This may suggest that comparing distributions based on the Wasserstein distance is a strategy that is bound to suffer from the so-called curse of dimensionality. Nevertheless, such theoretical performance in terms of rates of statistical convergence seems to be incompatible with the popularity of the Wasserstein distance based on the empirical performance observed in the previously mentioned application areas.

Our goal in this paper is to shed light on some of the fundamental reasons that explain the empirical performance of the Wasserstein distance as an effective way to compare distributions, guided by the following intuition. The Wasserstein distance (using, say, the Euclidean metric in $\mathbb{R}^d$) has substantial power to "separate" two distributions based on a wide and detailed range of characteristics. Meanwhile, some users of Wasserstein distances may be interested in only a subset of these characteristics (maybe a large subset, but just a subset, nonetheless). Hence, in the end, these users may be interested in only testing if an empirical sample is compatible with a subset of characteristics. Since this subset of characteristics of interest are likely to change from user to user or from task to task, the power of the Wasserstein distance to discriminate widely makes is particularly convenient for multiple users or tasks with different preferences because of this type of versatility. In practice, however, when testing if the data is compatible with the characteristics required for a particular user or task, such a user typically exploits the Wasserstein distance to obtain key insights and a deeper understanding while, in the end, making final decisions with a criterion that may ignore a lack of fit of certain aspects.

To be more precise, consider as a canonical example the process of using the Wasserstein distance in the Wasserstein GAN application [1]. The general goal is to fine tune a neural network to generate synthetic data that is similar in some sense to a target data set. The network is trained in order to minimize the Wasserstein distance. However, if the generative models eventually produce the desirable features (e.g., faces that appear to be realistic), we may choose to ignore imperfections in, for example, the background of the picture. Hence, "faces" are what we choose to emphasize in the training process and the rest of the data characteristics are not given as much importance.

The idea of choosing a hypothesis class corresponds precisely to modeling the set of characteristics that are important. The hypothesis class partitions the set of distributions into equivalence classes, where two distributions are equivalent if the expectations coincide over the hypothesis class. Formally, we posit that many users of the Wasserstein distance are actually testing if the data belongs to a certain equivalence class. To provide a solid statistical footing for such scenarios, this then involves computing the distance between the empirical measure and the target equivalence class, evaluating a corresponding asymptotic quantile statistic, and rejecting the hypothesis of membership in the target equivalence class for large values of the statistic relative to the desired confidence quantile.

More formally, suppose that $P_n$ denotes the empirical measure of independent and identically distributed (i.i.d.) samples $\{X_i\}_{i=1}^n \subseteq \mathbb{R}^d$ generated from a distribution $P_*$. Let us write $\mathcal{W}(P, P_n)$ to denote the Wasserstein distance [24] between $P_n$ and a given (Borel) probability measure $P$. (We recall the formal definition of the Wasserstein distance in Section 2.1.)

Let $\mathcal{B}$ be a given hypothesis class of interest. To avoid technicalities, let us focus in this introductory discussion on a given subset of the space of continuous and bounded functions with certain characteristics. Next, for $f \in \mathcal{B}$, we write $\mathbb{E}_P[f(X)]$ to denote the expectation of $f(X)$ under the measure $P$; so, for example, $\mathbb{E}_{P_n}[f(X)] = n^{-1} \sum_{i=1}^n f(X_i)$.

Our goal then in this paper is to study

$$R_n = \inf_P \{\mathcal{W}(P, P_n) : \mathbb{E}_P[f(X)] = \mathbb{E}_{P_*}[f(X)] \text{ for all } f \in \mathcal{B}\}. \qquad (1)$$

The main contributions of this paper are as follows. First, we provide a duality result that shows

$$R_n = \sup_{f \in \mathcal{LB}} \{\mathbb{E}_{P_*}[f(X)] - \mathbb{E}_{P_n}[f^c(X)]\},$$

where $f^c$ is a suitable transformation (to be described precisely in Theorem 1) and $\mathcal{LB}$ is the linear span generated by $\mathcal{B}$. If $\mathcal{B}$ is the class of all 1-Lipschitz functions and the cost function is the corresponding metric, then it turns out for $f \in \mathcal{B}$ that $f^c = f$ and we recover the celebrated Kanotorovich-Rubinstein duality.

The second contribution of this paper is to study the rate of statistical convergence for $R_n$. Note that, if $\mathcal{B}$ is the class of 1-Lipschitz functions, then $R_n$ will typically converge to zero at the rate

$O_p\left(n^{-1/(d\vee 2)}\right)$ [8], where $d$ is the dimension of the underlying space. If $\mathcal{B}$ is finite dimensional, then convergence of $R_n$ occurs at a parametric rate [3]. However, we also establish more general conditions that accommodate infinite dimensional hypothesis classes $\mathcal{B}$ and for which a parametric rate of convergence is also achievable, thus beating the curse of dimensionality. Examples of infinite-dimensional hypothesis classes, informed by a complex correlation structure, are considered.

Moreover, we are able to explicitly characterize the asymptotically limiting distribution of $n \times R_n$ as $n \to \infty$, which is the maximum of a Gaussian process, indexed by functions in $\mathcal{LB}$. The existence of this asymptotic distribution is critically important from the standpoint of using these results for hypothesis testing, as we have discussed above, either by explicit evaluation of quantiles or by means of subsampling as considered in [17].

There is a rapidly growing research literature discussing the statistical properties of the Wasserstein distance and how to beat the curse of dimensionality. Weed and Bach [25] claim that the Wasserstein distance enjoys a faster convergence rate if the true measure has support on a lower-dimensional manifold. Weed and Berthet [26] produce a new density estimator that converges faster if the true measure has sufficiently smooth density. Tameling et al. [20] recover the parametric rates of convergence, but under the assumption that the underlying measures are atomic. Genevay et al. [11] study Wasserstein distance with entropy regularization (Sinkhorn Divergences), but their convergence rate is exponential in the regularization power $\varepsilon$. In connection to our study, Blanchet et al. [2, 3] focus on finite hypothesis classes and prove that the canonical rate of statistical convergence can be obtained. We study cases in which the hypothesis class may form an infinite-dimensional vector space encoding complex information about the joint distribution, for which we are able to show, for the first time, that it is not only possible to also obtain a canonical rate of statistical convergence in these types of complex formulations, but to further obtain a characterization of the limiting distribution.

Our formulation is also related to distributionally robust optimization (DRO) with the Wasserstein distance metric [4, 16, 27, 3, 10, 5]. In this literature, estimators are obtained as the solution of a $\text{min-max}$ game in which the optimizer seeks to minimize a loss, while an adversary chooses a probability distribution inside a so-called "uncertainty set" defined around the empirical measure. The Wasserstein distance is used to describe the uncertainty set and $R_n$ is used to describe the radius of the uncertainty set (also called the size of uncertainty). One criterion for choosing the size of uncertainty is to minimize the size of a natural confidence region for the parameter of interest; refer to [3]. Under this criterion, it is shown that the optimal size of uncertainty coincides with a quantile of $R_n$ (which, in this literature, is known as the "Robust Wasserstein Profile" function).

The paper is organized as follows. In Section 2, we provide the necessary definitions and setup to state our duality result in compact spaces, which is presented in Section 2.2. Then, in Section 3, we discuss the statistical guarantee that $R_n$ satisfies, where we present a central limit theorem for $R_n$. Further, in Section 4, we extend our duality result and our statistical guarantee to non-compact spaces. Finally, Section 5 illustrates the use of our results in the context of a hypothesis testing example.

**Notation.** Let $\mathcal{C}^k(\Omega)$ represent the space of all $k$-th continuous differentiable functions defined on the domain $\Omega$, where $\mathcal{C}(\Omega)$ denotes the space of continuous functions and $\mathcal{C}_b(\Omega)$ the space of bounded continuous functions. Denote by $\mathcal{P}(\Omega)$ the space of all Borel probability measures on the underlying space $\Omega$. Let $L_1(\mu)$ be the space of all integrable functions with respect to measure $\mu$. Denote by $\mathbb{Z}_+$ the set of all positive integers and by $\|\cdot\|_F$ the Frobenius norm of a matrix. Let $\Rightarrow$ denote the weak convergence in a given probability space, and $\mathcal{N}(\mu, \sigma^2)$ a Gaussian distribution with mean $\mu$ and variance $\sigma^2$. For a vector $x \in \mathbb{R}^d$, we use $x^{(i)}, i = 1, 2, \ldots, d$, to denote the $i$-th entry of $x$.

## 2 Main Duality Result

The goal of this section is to present our new strong duality result, also providing the necessary definitions to do so. Recall that this result extends the existing optimal transport duality theory in a geometric sense by closing the gap between the renowned Kantorovich-Rubinstein duality result [24] at one extreme and the recent strong duality result in [3] at the other extreme. In doing so, our new strong duality result helps to reduce the computational burdens encountered in practice by establishing an equivalence with a problem that is easier and more computationally efficient to solve.

We start by reviewing the definition of the Wasserstein distance and the elements required to pose the dual problem. We then state our new strong duality result together with some examples of applying

the result, which further illustrate some of the benefits of our duality result. This is then followed by an extension of our strong duality result beyond the Wasserstein distance in (1) to the max-sliced Wasserstein distance in [7].

## 2.1 Wasserstein Distance

For a given closed set $\Omega \subseteq \mathbb{R}^d$, we endow $\Omega$ with a metric, denoted by $\varrho(\cdot)$, which may be naturally defined in terms of a norm such as $\varrho(x, y) = \|y - x\|$. Let $c : \Omega \times \Omega \to [0, \infty)$ be a continuous function with respect to $\varrho(\cdot)$. Then the optimal transport cost between $P, Q \in \mathcal{P}(\Omega)$ is defined as

$$
\begin{aligned}
\mathcal{D}_c(P, Q) \quad = \quad & \min_{\pi \in \mathcal{P}(\Omega \times \Omega)} \left\{ \left( \int c(x, w) \pi\, (\mathrm{d}x, \mathrm{d}w) \right) \right. \\
& \left. : \int_{w \in \mathbb{R}^d} \pi\, (\mathrm{d}x, \mathrm{d}w) = P\, (\mathrm{d}x)\, , \int_{x \in \mathbb{R}^d} \pi\, (\mathrm{d}x, \mathrm{d}w) = Q\, (\mathrm{d}w) \right\}.
\end{aligned}
$$

If $c(\cdot) = \varrho(\cdot)$, then $\mathcal{W}_1(P, Q) = \mathcal{D}_\varrho(P, Q)$ is the Wasserstein distance generated by such a metric [24]. However, we may also be interested in cases where $c(\cdot) = \varrho^r(\cdot)$ for $r > 1$ in order to study the Wasserstein distance of order $r$, which is defined as $\mathcal{W}_r(P, Q) = \mathcal{D}_{\varrho^r}^{1/r}(P, Q)$.

## 2.2 Strong Duality

The hypothesis class $\mathcal{B}(\Omega)$ is assumed to be given throughout our discussion which follows where we further assume that $\mathcal{B}(\Omega) \subseteq \mathcal{C}(\Omega) \cap L_1(P_*)$ for a targeting probability measure $P_*$. We may also assume, without loss of generality, that $\mathbf{1} \in \mathcal{B}(\Omega)$ (i.e., constant functions belong to the hypothesis class). Let $\mathcal{LB}(\Omega)$ denote the linear span generated by $\mathcal{B}(\Omega)$, namely

$$
\mathcal{LB}(\Omega) = \left\{ f(\cdot) = \sum_{i=1}^m \lambda_i f_i(\cdot) : \{f_i(\cdot)\}_{i=1}^m \subset \mathcal{B}(\Omega), \lambda \in \mathbb{R}^m, \text{ and } m \in \mathbb{Z}_+ \right\}.
$$

We formally state our assumptions as follows.

**Assumption 1.** *1. The function class satisfies $\mathcal{B}(\Omega) \subseteq \mathcal{C}(\Omega) \cap L_1(P_*)$.*

*2. The cost function $c(\cdot, \cdot)$ is a non-negative continuous function with $c(x, x) = 0$, for $x \in \Omega$.*

Given a probability measure $P_0 \in \mathcal{P}(\Omega)$ (which eventually will be taken as an empirical measure), we are interested in studying the robust Wasserstein profile function

$$
R_0 = \inf_{P \in \mathcal{P}(\Omega)} \{\mathcal{D}_c\, (P, P_0) : \mathbb{E}_P\, [f\, (X)] = \mathbb{E}_{P_*}\, [f\, (X)]\, , \text{ for all } f \in \mathcal{B}\, (\Omega)\}. \tag{2}
$$

Observe that writing $\mathcal{B}(\Omega)$ or $\mathcal{LB}(\Omega)$ in the definition of $R_0$ leads to an equivalent formulation due to the linearity of the constraints defining $R_0$. We now state our main duality result.

**Theorem 1.** *Suppose Assumption 1 is enforced and $\mathcal{B}(\Omega) \subset L_1(P_0)$. We then have the weak duality*

$$
R_0 \geq \sup_{f \in \mathcal{LB}(\Omega)} \{\mathbb{E}_{P_*}\, [f(X)] - \mathbb{E}_{P_0}\, [f^c\, (X)]\}\, ,
$$

*where $f^c$ is the c-transform of $f$, which is defined by*

$$
f^c(x) = \sup_{z \in \Omega} \{f(z) - c(z, x)\}\, .
$$

*Furthermore, if $\Omega$ is compact, we have the strong duality*

$$
R_0 = \sup_{f \in \mathcal{LB}(\Omega)} \{\mathbb{E}_{P_*}\, [f(X)] - \mathbb{E}_{P_0}\, [f^c\, (X)]\}\, . \tag{3}
$$

The key to the proof is first writing $R_0$ in a Lagrangian form and then applying Sion's minimax theorem [19]. The technical details and complete proof are provided in Appendix A.1.

**Remark 1.** *Notice that, for the strong duality, we require the sample space to be compact. For the non-compact space, the strong duality does not hold in general and should be treated on a case-by-case basis. We will discuss such strong duality results for some examples in Section 4.*

**Remark 2.** *Note that the dual formulation* (3) *shares some similarities with the Integral Probability Metric (IPM), which is defined as*

$$\mathrm{IPM}_{\mathcal{F}}(P, P_0) = \sup_{f \in \mathcal{F}} \left| \int f \mathrm{d}P - \int f \mathrm{d}P_0 \right|,$$

*for a function class $\mathcal{F}$. The similarities are not surprising since the dual formulations of Wasserstein distances have deep connections with IPM. However, it is important to note that our primary intention is not to define a new metric. Rather we seek to provide a thorough analysis of the Wasserstein distance, which has been the focus of a great deal of attention in the statistical learning research literature. In particular, we add a new modeling feature, which is the hypothesis class or the actor critic class. This induces a class of dual functions; and we note that our expression for the strong duality (generalizing the celebrated Kantorovich-Rubinstein duality) uses the combination of both the function $f$ and its $c$-transform $f^c$ in contrast with IPM.*

Problem (2) is an infinite-dimensional optimization problem that cannot be solved directly. Our main duality results (Theorem 1) enable us to compute $R_0$ using function approximators for functions in $\mathcal{LB}(\Omega)$, such as wavelet basis expansions. We will discuss computing $R_0$ in Section 5.

For now, let us consider a few examples that apply our results to illustrate some of the benefits which they provide. In order to connect these examples with our future statistical development, recall that $\{X_i\}_{i=1}^{n} \subseteq \Omega$ are i.i.d. samples from a data-generating distribution $P_* \in \mathcal{P}(\Omega)$ and that $P_n = \frac{1}{n} \sum_{i=1}^{n} \delta_{X_i}$ is the corresponding empirical measure. We next apply our strong duality result where $P_0$ is replaced by $P_n$ and the corresponding $R_n$ is defined as

$$R_n = \inf_{P \in \mathcal{P}(\Omega)} \{\mathcal{D}_c(P, P_n) : \mathbb{E}_P[f(X)] = \mathbb{E}_{P_*}[f(X)], \text{ for all } f \in \mathcal{B}(\Omega)\}.$$

**Example 1.** *Suppose $\mathcal{LB}(\Omega)$ is sufficiently rich to uniquely determine any distributions and assume that $c = \varrho$. Then, we might assume that $\mathcal{LB}(\Omega)$ is the space of all Lipschitz functions, which also determines any distribution. Let $\mathrm{Lip}_1(\Omega)$ be the space of all 1-Lipschitz functions. Hence, by our weak duality result, we have*

$$\sup_{f \in \mathrm{Lip}_1(\Omega)} \{\mathbb{E}_{P_*}[f(X)] - \mathbb{E}_{P_n}[f^c(X)]\} \leq R_n.$$

*On the other hand, since $f(x) \leq \sup_{z \in \Omega} \{f(z) - c(z, x)\}$, we also have*

$$R_n \leq \sup_{f \in \mathcal{LB}(\Omega)} \{\mathbb{E}_{P_*}[f^c(X)] - \mathbb{E}_{P_n}[f^c(X)]\}.$$

*Finally, it is well known (see, e.g., [24]) that $f^c(x)$ is a 1-Lipschitz function, and therefore*

$$\sup_{f \in \mathcal{LB}(\Omega)} \{\mathbb{E}_{P_*}[f^c(X)] - \mathbb{E}_{P_n}[f^c(X)]\} \leq \sup_{f^c \in \mathrm{Lip}_1(\Omega)} \{\mathbb{E}_{P_*}[f^c(X)] - \mathbb{E}_{P_n}[f^c(X)]\}.$$

*Consequently, if $\mathcal{LB}(\Omega)$ determines any distribution, then our result recovers the renowned Kantorovich-Rubinstein duality result [24, Theorem 5.10]:*

$$R_n = \sup_{f \in \mathrm{Lip}_1(\Omega)} \{\mathbb{E}_{P_*}[f(X)] - \mathbb{E}_{P_n}[f(X)]\} = \mathcal{W}_1(P_*, P_n).$$

*It is important to keep in mind that, if $P_*$ has bounded moments, then $R_n = O\left(n^{-1/(d \vee 2)}\right)$ as $n \to \infty$ (see, e.g., [8, Theorem 1]).*

**Example 2.** *Suppose that $\mathcal{B}(\Omega)$ is finite dimensional, such as $\mathcal{B}(\Omega) = \{f_i(x)\}_{i=1}^{K}$. Then, we have*

$$R_n = \sup_{\lambda \in \mathbb{R}^K} \left\{ \mathbb{E}_{P_*} \left[ \sum_{i=1}^{K} \lambda_i f_i(X) \right] - \mathbb{E}_{P_n} \left[ \sup_{z \in \Omega} \left\{ \sum_{i=1}^{K} \lambda_i f_i(z) - c(z, X) \right\} \right] \right\},$$

*which recovers the duality result obtained in [3]. Note that [3] also provides a typical rate $R_n = O_p\left(n^{-1}\right)$ as $n \to \infty$ under some regularity conditions.*

**Example 3.** *Fix linearly independent unit vectors $\theta_1, \ldots, \theta_K \in \mathbb{R}^d$, $K \leq d$, and let a function class $\mathcal{F}_{\mathcal{B}} \subseteq \mathcal{C}_b(\mathbb{R})$ collect some bounded continuous functions in $\mathbb{R}$. We consider the function class $\mathcal{B}(\Omega) = \cup_{i=1}^{K} \mathcal{B}_i(\Omega)$, where $\mathcal{B}_i(\Omega) = \left\{ f(\theta_i^\top \cdot)|_\Omega : f \in \mathcal{F}_{\mathcal{B}} \right\}$, in which case*

$$\mathcal{LB}(\Omega) = \left\{ f(\cdot) = \sum_{i=1}^{K} \lambda_i f_i(\theta_i^\top \cdot)|_\Omega : \{f_i(\cdot)\}_{i=1}^{K} \subset \mathcal{F}_{\mathcal{B}}, \lambda \in \mathbb{R}^K \right\}.$$

*This example is particularly interesting because it is infinite dimensional if $\mathcal{F}_{\mathcal{B}}$ is infinite dimensional. The hypothesis class carries a substantial amount of information about the dependence structure of $P_*$ and yet, as we shall see, for this hypothesis class and the cost function $c(x,y) = \|x - y\|_2^2$, we also conclude that $R_n = O_p\left(n^{-1/2}\right)$ for $\Omega = \mathbb{R}^d$ (Theorem 4 below) and $R_n = O_p\left(n^{-1}\right)$ under suitable regularity (Theorem 2 below).*

At first glance, Example 3 is similar to the max-sliced Wasserstein distance [7]. Recall that the max-sliced Wasserstein distance is defined as

$$\text{max-}\mathcal{W}_r\left(P, Q\right) = \left[\max_{\theta:\|\theta\|_2=1} \mathcal{W}_r\left(\theta_\sharp P, \theta_\sharp Q\right)^r\right]^{1/r},$$

where $\theta_\sharp P(\theta_\sharp Q)$ is the push-forward measure from $\mathcal{P}(\Omega)$ to $\mathcal{P}\left(\theta^\top \Omega\right)$ such that, for any Borel set $A$ in $\theta^\top \Omega$,

$$\left(\theta_\sharp P\right)(A) = P\left(\{x \in \Omega : \theta^\top x \in A\}\right). \tag{4}$$

Proposition 1 provides a strong duality result for $\text{max-}\mathcal{W}_r\left(P, Q\right)$.

**Proposition 1.** *Consider $\Omega = \mathbb{R}^d, r = 2$, and $\varrho(x,y) = |x - y|$, for $x, y \in \mathbb{R}$. Denote by $S^{d-1}$ a unit sphere in $\mathbb{R}^d$, i.e., $S^{d-1} = \left\{x \in \mathbb{R}^d : \|x\|_2 = 1\right\}$. Then, for $\Theta \subset S^{d-1}$, we have the strong duality*

$$\max_{\theta \in \Theta} \mathcal{W}_2\left(\theta_\sharp P, \theta_\sharp Q\right)^2 = \sup_{f \in \mathcal{B}_{\max}(\mathbb{R}^d, \Theta)} \left\{\mathbb{E}_P\left[f\left(X\right)\right] - \mathbb{E}_Q\left[f^c(X)\right]\right\}, \tag{5}$$

*where the cost function $c(x,y) = \|x - y\|_2^2$ and*

$$\mathcal{B}_{\max}\left(\mathbb{R}^d, \Theta\right) = \left\{f\left(\theta^\top \cdot\right) : f \in \mathcal{C}_b(\mathbb{R}), \theta \in \Theta\right\}.$$

*In particular, for the max-sliced distance, we have the strong duality*

$$\left(\text{max-}\mathcal{W}_2\left(P, Q\right)\right)^2 = \sup_{f \in \mathcal{B}_{\max}(\mathbb{R}^d, S^{d-1})} \left\{\mathbb{E}_P\left[f\left(X\right)\right] - \mathbb{E}_Q\left[f^c(X)\right]\right\}.$$

The proof of Proposition 1 is provided in Appendix A.2. The key difference between the dualities (1) and (5) is that $\mathcal{B}_{\max}\left(\mathbb{R}^d, \Theta\right)$ is not a vector space in general. Therefore, even if $\Theta = \{\theta_i\}_{i=1}^K$, $\mathcal{LB}(\mathbb{R}^d)$ could be much larger than $\mathcal{B}_{\max}\left(\mathbb{R}^d, \Theta\right)$.

## 3 Statistical Convergence

In this section, we present a formal central limit theorem result on the rate of statistical convergence for $R_n$ in the case of infinite dimensional constraints, which also extends and improves the corresponding results in [3] for the finite dimensional case. This further extends conventional results on the rate of statistical convergence for Wasserstein distances between an empirical distribution and the true (unknown) distribution. Such central limit theorem results on the rate of statistical convergence for $R_n$ provide a critically important understanding that can inform and guide algorithms, computation, and experiments.

Following the setting in Example 3, we consider a convex compact domain $\Omega$ and let $\mathcal{B}_i(\Omega)$ be any subclass of the function class $\left\{f(\theta_i^\top \cdot)|_\Omega : f \in \mathcal{C}^2(\mathbb{R})\right\}$. As an analog of $\mathcal{LB}(\Omega)$, we define $\mathcal{LB}_i(\Omega)$ to be

$$\mathcal{LB}_i(\Omega) = \left\{f(\cdot) = \sum_{j=1}^m \lambda_j f_j(\cdot) : \{f_j(\cdot)\}_{j=1}^m \subset \mathcal{B}_i(\Omega), \lambda \in \mathbb{R}^m, \text{ and } m \in \mathbb{Z}_+\right\}.$$

Notice that any function in $\mathcal{LB}_i(\Omega)$ can be written as $f(\theta_i^\top x)$. We assume that the function classes $\mathcal{LB}_i(\Omega)$ satisfy the following condition.

**Assumption 2.** *For any $f\left(\theta_i^\top \cdot\right) \in \mathcal{LB}_i(\Omega), i = 1, 2, \ldots, K$, the ratio bound*

$$\frac{\sup_{x \in \Omega}\left|f''(\theta_i^\top x)\right|}{\sqrt{\int_\Omega f'(\theta_i^\top x)^2 \mathrm{d}x}} \leq M$$

*holds for a universal constant $M \in (0, +\infty)$, where we use the convention $0/0 = 0$.*

As in Example 3, we consider the function space $\mathcal{B}(\Omega) = \cup_{i=1}^K \mathcal{B}_i(\Omega)$. We make further assumptions on the domain $\Omega$, the data-generating probability measure $P_*$, and the linear projection vectors $\theta_1, \theta_2, \ldots, \theta_K$ in Assumption 3 as follows.

**Assumption 3.** *1. The sample space $\Omega$ is a convex and compact subset of $\mathbb{R}^d$.*

*2. The data-generating probability measure $P_*$ has a non-zero density $f_{P_*}$ with respect to Lebesgue measure in $\mathbb{R}^d$. The density has a uniform non-zero lower bound, i.e., $f_{P_*}(x) \geq \underline{b} > 0$ for $x \in \Omega$.*

*3. The vectors $\theta_1, \ldots, \theta_K$ are linearly independent with $\|\theta_i\|_2 = 1$ for $i = 1, 2, \ldots, K$.*

**Theorem 2.** *Suppose Assumptions 1, 2 and 3 are enforced. For the cost function $c(x, y) = \|x - y\|_2^2$, we then have the central limit theorem result*

$$nR_n \Rightarrow \sup_{f \in \mathcal{LB}(\Omega)} \left\{ -2H^f - \mathbb{E}_{P_*} \left[ \|\nabla_X f(X)\|_2^2 \right] \right\},$$

*where $\nabla_x f(x)$ is the gradient of $f(\cdot)$ evaluated at $x$ and $H^f$ is a Gaussian process indexed by $f$ with*

$$H^f \sim \mathcal{N}(0, \mathrm{var}\,(f(X))) \text{ and } \mathrm{cov}(H^{f_1}, H^{f_2}) = \mathrm{cov}\,(f_1(X), f_2(X)).$$

*Sketch of Proof.* Define $\mathcal{UB}(\Omega) = \left\{ f(\cdot) \in \mathcal{LB}(\Omega) : \mathbb{E}_{P_*} \left[ \|\nabla_x f(X)\|_2^2 \right] = 1, f(0) = 0 \right\}$. By Theorem 1, we have $nR_n$ is equal to

$$\sup_{\lambda \in \mathbb{R}} \sup_{f \in \mathcal{UB}(\Omega)} \left\{ -2\lambda H_n^f - \frac{1}{n} \sum_{i=1}^n \left[ \sup_{X_i + \Delta/\sqrt{n} \in \Omega} \left\{ 2\lambda\sqrt{n} \left( f\left(X_i + \Delta/\sqrt{n}\right) - f(X_i)\right) - \|\Delta\|_2^2 \right\} \right] \right\},$$

where $H_n^f = n^{-1/2} \left( \sum_{i=1}^n f(X_i) - E_P[f(X)] \right)$. Then, by the uniform convergence theory of the $P$-Donsker class and the $P$-Glivenko-Cantelli class (see [21, Chapter 19]), we obtain for any $b > 0$

$$\sup_{|\lambda| \leq b} \sup_{f \in \mathcal{UB}(\Omega)} \left\{ \lambda H_n^f \right\} \Rightarrow \sup_{|\lambda| \leq b} \sup_{f \in \mathcal{UB}(\Omega)} \left\{ \lambda H^f \right\}, \text{ and} \tag{6}$$

$$\sup_{|\lambda| \leq b} \sup_{f \in \mathcal{UB}(\Omega)} \left| \frac{1}{n} \sum_{i=1}^n \left[ \sup_{X_i + \Delta/\sqrt{n} \in \Omega} \left\{ 2\lambda\sqrt{n} \left( f(X_i + \Delta/\sqrt{n}) - f(X_i)\right) - \|\Delta\|_2^2 \right\} \right] - \lambda^2 \right| \to 0. \tag{7}$$

Furthermore, we show that $\lambda$ is bounded with high probability when $n$ is large. Upon combining (6) and (7) with the boundedness of $\lambda$, we obtain the desired central limit theorem. $\qquad\square$

Theorem 2 demonstrates a parametric rate of convergence, in contrast with the standard $O\left(n^{-1/(d\vee2)}\right)$ convergence rate of Wasserstein distances (see, e.g., [8, Theorem 1]). The technical details and complete proof are presented in Appendix A.3.

## 4 Extension to Non-Compact Spaces

Our previous discussions and results on strong duality and statistical convergence have been limited to the case of compact domains. We now turn to consider results on strong duality and statistical convergence for the case when the sample space $\Omega$ is not compact.

We start by considering our results on strong duality in the case of non-compact domains, and then considering our results on the rate of statistical convergence in the case of non-compact domains, both following along the lines of Example 3 above.

**Theorem 3.** *Consider $\Omega = \mathbb{R}^d$ and a continuous cost function $c(\cdot, \cdot) : \mathbb{R}^d \times \mathbb{R}^d \to [0, \infty)$ with $c(x, x) = 0$. Assume that $\mathbb{E}_{P_*}[c(X, y)] < \infty$ for any $y \in \mathbb{R}^d$, and that the set $\left\{ x \in \mathbb{R}^d : c(x, x_0) \leq a \right\}$ is compact for any $a > 0$. Following the setting in Example 3, for linearly independent unit vectors $\theta_1, \ldots, \theta_K$ and $\mathcal{F}_B = \mathcal{C}_b(\mathbb{R})$, we have the strong duality*

$$R_n = \sup_{f \in \mathcal{LB}(\mathbb{R}^d)} \left\{ \mathbb{E}_{P_*}[f(X)] - \mathbb{E}_{P_n}[f^c(X)] \right\}.$$

*Sketch of proof.* Since we have the weak duality proven in Theorem 1, we only need to show

$$\mathcal{D} := \sup_{f \in \mathcal{LB}(\mathbb{R}^d)} \{\mathbb{E}_{P_*}[f(X)] - \mathbb{E}_{P_n}[f^c(X)]\} \geq R_n.$$

Our strategy for this proof is to pick a series of large compact sets, so that we can approximate the solution to the primal problem by restricting the functions $c(\cdot, \cdot)$ and $f$ on the compact set.

We then apply strong duality for the compact problem and subsequently show that the dual optimal value $\mathcal{D}$ can be approximated by the dual optimal value of the compact problem, when we apply the truncation to the cost function $c_a(x, y) = \min\{a, c(x, y)\}$. Finally, the optimal value with the cost function $c_a(x, y)$ converges to the optimal value with the cost function $c(x, y)$. □

The detailed proof of Theorem 3 is provided in Appendix A.4. An important element which distinguishes the proof of the results from standard strong duality in optimal transport is that the usual technique to construct improving dual functions is not applicable since $f^c \notin \mathcal{LB}(\mathbb{R}^d)$ in general.

We next study the rate of statistical convergence within the context of Example 3.

**Theorem 4.** *Assume* $\Omega = \mathbb{R}^d$ *and the cost function* $c(x, y) = \|x - y\|_2^2$ *with* $\mathbb{E}_{P_*}[\|X\|_2^{4+\epsilon}] < \infty$ *for some* $\epsilon > 0$. *Let* $M(P_*) = \max\{1, \mathbb{E}_{P_*}[\|X\|_2^{4+\epsilon}]\}$. *Following the setting in Example 3, for linearly independent unit vectors* $\theta_1, ..., \theta_K$ *and any* $\mathcal{F}_\mathcal{B} \subset \mathcal{C}_b(\mathbb{R})$, *there exists a universal constant* $C$ *such that* $\mathbb{E}[R_n] \leq C\rho^* K(M(P_*))^2 n^{-1/2}$, *where* $\rho^*$ *denotes the spectral radius of the matrix* $C_K = [\theta_1, \theta_2, \ldots, \theta_K]^\top$.

The key to the proof is to perform the transformation $Y_K = C_K X$ and to apply the standard convergence result in [8, Theorem 1]. The technical details and complete proof are provided in Appendix A.5.

**Remark 3.** *The convergence rate* $O_p(1/\sqrt{n})$ *in Theorem 4 is slower than the rate* $O_p(1/n)$ *in Theorem 2. We emphasize that the rate* $O_p(1/\sqrt{n})$ *is also tight in situations where the support is non-compact. It is consistent with the observation in the one-dimensional Wasserstein distance of order* 2 *[6, Corollary 5.10].*

## 5 Numerical Experiments

We provide experimental results on testing the hypothesis that a set of $n$ samples $\{X_1, X_2, \ldots, X_n\} \subset \mathbb{R}^d$ is compatible with a candidate distribution $P_*$ for a set of user-desired characteristics, specifically the test described in Example 3. The projection directions $\{\theta_1, \ldots, \theta_K\}$ could be viewed as the characteristics of interest to the user (as discussed in the introduction). Theorem 4 shows that, if the hypothesis is true, then the robust Wasserstein profile function $R_n = O_p(n^{-1/2})$. We implement the test by first estimating this distribution of $R_n$ in its dual form (3). The hypothesis test can then be conducted in a standard manner by constructing the test statistic $R_n$ for the given empirical distribution $P_n = n^{-1} \sum_{i=1}^n \delta_{X_i}$ and checking whether it is within the desired confidence level.

The key step in estimating $R_n$ is to solve for $f^c(x)$. Let $\Omega = \mathbb{R}^d$, $C_K = [\theta_1, \theta_2, \ldots, \theta_K]^\top$ and $\Gamma_K = C_K C_K^\top$. We then have $f^c(x) = \sup_{z \in \mathbb{R}^K} \left( \sum_{j=1}^K f_j(\theta_j^\top x + z^{(j)}) - z^\top \Gamma_K^{-1} z \right)$, referring to Appendix B.2 for the technical details. Therefore, the inner supremum is a $K$-dimensional optimization problem instead of a $d$-dimensional problem.

We use Marr wavelet basis functions [13] to approximate the function class $\mathcal{B}(\Omega)$. In particular, we use a finite collection $\{b_l\}_{l=1}^L$ of Marr wavelet bases, where we provide the explicit expressions in Appendix B.1. Hence, the $R_n$ is approximated by :

$$\hat{R}_n = \sup_{w_{jl}} \left\{ \mathbb{E}_{P_*}\left[\sum_{j=1}^K \sum_{l=1}^L w_{jl} b_l(\theta_j^\top X)\right] - \frac{1}{n}\sum_{i=1}^n \sup_{z_i} \sum_{j=i}^K \sum_{l=1}^L \left[w_{jl} b_l(\theta_j^\top x_i + z_i^{(j)}) - z_i^\top \Gamma_k^{-1} z_i\right] \right\}.$$

Stochastic approximation (SA, also known as SGD) iterations are used to obtain the optimal solution of $\hat{R}_n$. Specifically, each SA iteration estimates expectations $\mathbb{E}_{P_*}$ using a mini-batch sample from $P_*$ of size 50. During each iteration, the $n$ inner supremum problems are solved by Newton iterations with 150 restarts (see Appendix B.5 for the details).

To reject the hypothesis that the given set is from $P_*$, we use the 95% quantile of the distribution of $\hat{R}_n$ obtained when the empirical sets are indeed from $P_*$ as a threshold. We construct an estimate of this quantile from the empirical distribution of $\hat{R}_n$ obtained by from 50 instances of $n$ sized samples $P_n$ generated from $P_*$. The $P_*$ distribution is an equal mixture of four standard Gaussians with $d = 20$. Our $n$-sized test set $P_n^{\text{alt}}$ is from an alternate distribution $P_n^{\text{alt}}$ that is also a mixture of standard Gaussians but with different centering points. The test statistic $\hat{R}_n^{\text{alt}}$ computed for $P_n^{\text{alt}}$ against $P_*$ is thus tested against the 95% quantile of $\hat{R}_n$ to decide on the hypothesis that $P_n^{\text{alt}}$ is from $P_*$. Three ($K = 3$) projection directions $\theta_j$ are carefully chosen to be linearly independent and such that they can reveal the modes of $P_*$, the user-preferred characteristics of interest. We set $n = 25$ and choose $L \sim 30$ basis functions. Each computational run to estimate $\hat{R}_n(\hat{R}_n^{\text{alt}})$ for a given $P_n(P_n^{\text{alt}})$ takes on average 10 minutes to compute on a dual AMD EPYC 7301 16-Core Processor machine with 64GB of memory utilizing 50 subprocesses to solve the inner supremum problems in parallel.

We report the results in Figure 1. On the left we plot the projection of the two distributions $P_*$ (blue shade) and $P_*^{\text{alt}}$ (red shade) on the three $\theta_j$ directions. Notice that each plot reveals that $P_*$ has at least three modes along the projection direction $\theta_j$; while, on the other hand, these directions $\theta_j$ reveal only one mode each for $P_*^{\text{alt}}$. The right plot of Figure 1 shows the distributions of $\hat{R}_n$ (blue histogram) and $\hat{R}_n^{\text{alt}}$ (red histogram) estimated by computing for $\hat{R}_n$ and $\hat{R}_n^{\text{alt}}$ repeatedly for 50 times each. The black dashed line marks the estimated 95% quantile of $\hat{R}_n$. In this case, we control the type I error as 5% and obtain a type II error as 32%. This shows that the method, based on our theoretical results, can efficiently distinguish between two $d = 20$ distributions in terms of the user-preferred characteristics while providing good accuracy even for relatively small values of $n$.

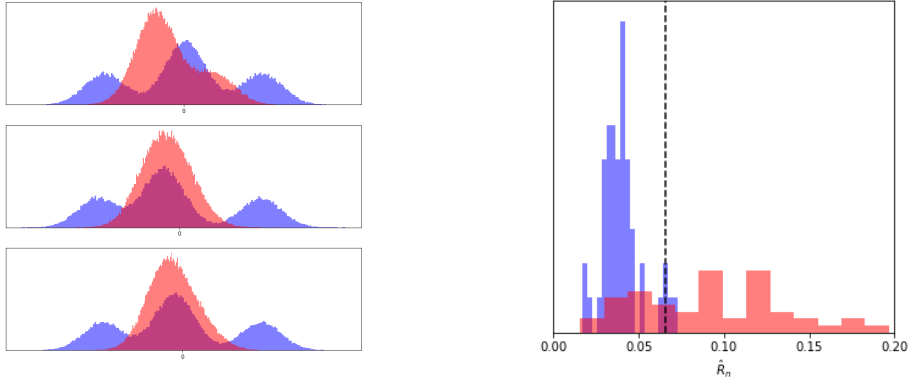

Figure 1: Left: projections of $P^*$ (blue shade) and $P_*^{\text{alt}}$ (red shade) along the three $\theta_j$ directions; Right: histograms of 50 samples of $\hat{R}_n$ (blue histogram) and $\hat{R}_n^{\text{alt}}$ (red histogram) with the 95% quantile of $\hat{R}_n$ marked as a dashed black line.

## 6    Discussion

Motivated by the intuition that decision makers may only be concerned with some characteristics instead of all the details of the entire distribution, we consider the problem of projecting the empirical measure under the Wasserstein distance to a set of probability measures that are constrained to satisfy a family of expectations over a class of functions. In particular, we study theoretical aspects of the robust Wasserstein profile functions $R_n$. We believe this work provides important insights into the empirical success of the Wasserstein distance despite the curse of dimensionality. Interesting future directions include studying statistical convergence for general function classes, developing efficient algorithms to compute $R_n$, and applying our methods in practice leveraging our theoretical insights.

## Acknowledgement

Material in this paper is based upon work supported by the Air Force Office of Scientific Research under award number FA9550-20-1-0397. Additional support is gratefully acknowledged from NSF grants 1915967, 1820942 and 1838576.

## Broader Impact

This is a theoretical contribution that, nevertheless, has the potential of impacting a wide range of application domains in business, engineering and science. In particular, all of those in which the Wasserstein distance has been extensively used as a statistical inference tool (e.g. image analysis and computer vision, signal processing, operations research, and so on). Because our paper provides a step towards breaking the curse of dimensionality in statistical rates of convergence, we believe that we have the potential of enabling more applications to multiple hypothesis testing (e.g., certifying Wasserstein GANs). In turn, we plan to improve human resource development by including some of the main findings in this paper in Ph.D. courses.

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
