[Supplementary Material]

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

\left(\mathrm{d}x, \mathrm{d}w\right) = P\left(\mathrm{d}x\right), \int_{x \in \mathbb{R}^d} \pi\left(\mathrm{d}x, \mathrm{d}w\right) = Q\left(\mathrm{d}w\right) \right\}.
\end{aligned}
$$

If $c\left(\cdot\right) = \varrho\left(\cdot\right)$, then $\mathcal{W}_1(P, Q) = \mathcal{D}_\varrho(P, Q)$ is the Wasserstein distance generated by such a metric [24]. However, we may also be interested in cases where $c\left(\cdot\right) = \varrho^r\left(\cdot\right)$ for $r > 1$ in order to study the Wasserstein distance of order $r$, which is defined as $\mathcal{W}_r(P, Q) = \mathcal{D}_{\varrho^r}^{1/r}(P, Q)$.

## 2.2 Strong Duality

The hypothesis class $\mathcal{B}(\Omega)$ is assumed to be given throughout our discussion which follows where we further assume that $\mathcal{B}(\Omega) \subseteq \mathcal{C}(\Omega) \cap L_1\left(P_*\right)$ for a targeting probability measure $P_*$. We may also assume, without loss of generality, that $\mathbf{1} \in \mathcal{B}(\Omega)$ (i.e., constant functions belong to the hypothesis class). Let $\mathcal{LB}(\Omega)$ denote the linear span generated by $\mathcal{B}(\Omega)$, namely

$$
\mathcal{LB}(\Omega) = \left\{ f(\cdot) = \sum_{i=1}^m \lambda_i f_i(\cdot) : \{f_i(\cdot)\}_{i=1}^m \subset \mathcal{B}(\Omega), \lambda \in \mathbb{R}^m, \text{ and } m \in \mathbb{Z}_+ \right\}.
$$

We formally state our assumptions as follows.

**Assumption 1.** *1. The function class satisfies $\mathcal{B}(\Omega) \subseteq \mathcal{C}(\Omega) \cap L_1\left(P_*\right)$.*

*2. The cost function $c(\cdot, \cdot)$ is a non-negative continuous function with $c(x, x) = 0$, for $x \in \Omega$.*

Given a probability measure $P_0 \in \mathcal{P}(\Omega)$ (which eventually will be taken as an empirical measure), we are interested in studying the robust Wasserstein profile function

$$
R_0 = \inf_{P \in \mathcal{P}(\Omega)} \left\{ \mathcal{D}_c\left(P, P_0\right) : \mathbb{E}_P\left[f\left(X\right)\right] = \mathbb{E}_{P_*}\left[f\left(X\right)\right], \text{ for all } f \in \mathcal{B}\left(\Omega\right) \right\}. \tag{2}
$$

Observe that writing $\mathcal{B}\left(\Omega\right)$ or $\mathcal{LB}\left(\Omega\right)$ in the definition of $R_0$ leads to an equivalent formulation due to the linearity of the constraints defining $R_0$. We now state our main duality result.

**Theorem 1.** *Suppose Assumption 1 is enforced and $\mathcal{B}(\Omega) \subset L_1(P_0)$. We then have the weak duality*

$$
R_0 \geq \sup_{f \in \mathcal{LB}(\Omega)} \left\{ \mathbb{E}_{P_*}\left[f(X)\right] - \mathbb{E}_{P_0}\left[f^c\left(X\right)\right] \right\},
$$

*where $f^c$ is the c-transform of $f$, which is defined by*

$$
f^c(x) = \sup_{z \in \Omega} \left\{ f(z) - c(z, x) \right\}.
$$

*Furthermore, if $\Omega$ is compact, we have the strong duality*

$$
R_0 = \sup_{f \in \mathcal{LB}(\Omega)} \left\{ \mathbb{E}_{P_*}\left[f(X)\right] - \mathbb{E}_{P_0}\left[f^c\left(X\right)\right] \right\}. \

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

# Appendix A   Proofs of Main Results

## Appendix A.1   Proof of Theorem 1

Let $\mathcal{M}(\Omega), \mathcal{M}_+(\Omega)$ and $\mathcal{M}_+^a(\Omega)$ denote the set of all signed measures, positive Borel measures and positive Borel measures with total mass less than or equal to $a$ on the underlying space $\Omega$, respectively, where $\mathcal{M}(\Omega), \mathcal{M}_+(\Omega)$ and $\mathcal{M}_+^a(\Omega)$ are equipped with the weak topology. Recall that, in the weak topology, $\pi_n \Rightarrow \pi$ if and only if, for every continuous and bounded function $h : \Omega \to \mathbb{R}$, we have that $\int h \mathrm{d}\pi_n \to \int h \mathrm{d}\pi$ as $n \to \infty$.

We will prove some of our duality results in this section, where we divide the proof into two steps:

1. Prove the weak duality in general spaces.

2. Prove the strong duality in compact spaces.

Let us first rewrite $R_0$ as a linear programming problem:

$$R_0 = \inf_{\pi \in \mathcal{M}_+(\Omega \times \Omega), P \in \mathcal{M}_+(\Omega)} \int_{\Omega \times \Omega} c(x,y)\pi(\mathrm{d}x, \mathrm{d}y), \tag{A.1}$$

$$s.t. \quad \pi(A \times \Omega) = P_0(A), \pi(\Omega \times A) = P(A) \text{ for every measurable set } A,$$
$$\mathbb{E}_P[f(X)] = \mathbb{E}_{P_*}[f(X)] \text{ for all } f \in \mathcal{B}(\Omega).$$

## Appendix A.1.1   Weak Duality

We consider the set $\mathcal{JL}(c) = \{(\alpha, \beta) \in L_1(P_*) \times L_1(P_0) : \alpha(x) + \beta(y) \le c(x,y)\}$. Notice that

$$R_0 = \inf_{\pi \in \mathcal{M}_+(\Omega \times \Omega), P \in \mathcal{M}_+(\Omega)} \sup_{(\alpha,\beta) \in L_1(P_*) \times L_1(P_0), f \in \mathcal{LB}(\Omega)} \left\{ \int_{\Omega \times \Omega} c(x,y)\pi(\mathrm{d}x, \mathrm{d}y) + \right.$$

$$\left. (\mathbb{E}_{P_*}[f(X)] - \mathbb{E}_P[f(X)]) + \mathbb{E}_P[\alpha(X)] + \int_\Omega \beta(y)P_0(\mathrm{d}y) - \int_{\Omega \times \Omega} (\alpha(x) + \beta(y))\pi(\mathrm{d}x, \mathrm{d}y) \right\}.$$

Letting $\alpha(x) = f(x)$, we obtain

$$R_0 \ge \inf_{\pi \in \mathcal{M}_+(\Omega \times \Omega), P \in \mathcal{M}_+(\Omega)} \sup_{\beta \in \mathcal{C}_b(\Omega), f \in \mathcal{LB}(\Omega)} \mathbb{E}_{P_*}[f(X)]$$

$$+ \int_{\Omega \times \Omega} c(x,y)\mathrm{d}\pi(x,y) + \left[ \int_\Omega \beta(y)P_0(\mathrm{d}y) - \int_{\Omega \times \Omega} (f(x) + \beta(y))\pi(\mathrm{d}x, \mathrm{d}y) \right]$$

$$\ge \inf_{\pi \in \mathcal{M}_+(\Omega \times \Omega), P \in \mathcal{M}_+(\Omega)} \sup_{(f,\beta) \in \mathcal{JL}(c), f \in \mathcal{LB}(\Omega)} \mathbb{E}_{P_*}[f(X)]$$

$$+ \left( \int_{\Omega \times \Omega} c(x,y)\pi(\mathrm{d}x, \mathrm{d}y) - \int_{\Omega \times \Omega} (f(x) + \beta(y))\pi(\mathrm{d}x, \mathrm{d}y) \right) + \int_\Omega \beta(y)P_0(\mathrm{d}y)$$

$$\ge \inf_{\pi \in \mathcal{M}_+(\Omega \times \Omega), P \in \mathcal{M}_+(\Omega)} \sup_{(f,\beta) \in \mathcal{JL}(c), f \in \mathcal{LB}(\Omega)} \mathbb{E}_{P_*}[f(X)] + \int_\Omega \beta(y)P_0(\mathrm{d}y)$$

$$= \sup_{(f,\beta) \in \mathcal{JL}(c), f \in \mathcal{LB}(\Omega)} \mathbb{E}_{P_*}[f(X)] + \int_\Omega \beta(y)P_0(\mathrm{d}y).$$

It is readily verified that $f(x) - f^c(y) \le c(x,y)$ and $f^c(y)$ is lower semicontinuous (see, e.g., [24, Remark 5.5]), and thus measurable. Since $f^c(x) \ge f(x)$, we have $\mathbb{E}_{P_0}[f^c(X)] > -\infty$. Moreover, if $\mathbb{E}_{P_0}[f^c(X)] = +\infty$, then $\mathbb{E}_{P_*}[f(X)] - \mathbb{E}_{P_0}[f^c(X)] = -\infty$. By choosing $f = \beta = 0$, we obtain

$$\sup_{(f,\beta) \in \mathcal{JL}(c), f \in \mathcal{LB}(\Omega)} \mathbb{E}_{P_*}[f(X)] + \int_\Omega \beta(y)P_0(\mathrm{d}y) \ge 0.$$

If $f^c(\cdot) \notin L_1(P_0)$, then $f^c(\cdot)$ cannot be the optimizer, and therefore we have

$$\sup_{(f,\beta)\in\mathcal{JL}(c),f\in\mathcal{LB}(\Omega)} \left( \mathbb{E}_{P_*}[f(X)] + \int_\Omega \beta(y)P_0(\mathrm{d}y) \right) = \sup_{f\in\mathcal{LB}(\Omega)} \left( \mathbb{E}_{P_*}[f(X)] - \mathbb{E}_{P_0}[f^c(X)] \right).$$

This completes the weak duality proof.

### Appendix A.1.2 Strong Duality in Compact Spaces

We assume $\Omega$ is a compact space and consider the set $\mathcal{J}(c) = \{(\alpha,\beta) \in \mathcal{C}(\Omega) \times \mathcal{C}(\Omega) : \alpha(x) + \beta(y) \leq c(x,y)\}$. Notice that, in compact spaces, $\mathcal{C}(\Omega) = \mathcal{C}_b(\Omega)$ and thus $\mathcal{C}(\Omega) \in L_1(P)$ for any probability measure $P \in \mathcal{P}(\Omega)$. Sion's minimax Theorem [19], which will be useful for our proof, can be expressed as follows.

**Theorem A1** (Sion's minimax Theorem). *Consider two convex spaces $M$ and $N$, one of which is compact, and let $g : M \times N \to R$ be such that, for each $y \in N$, $g(\cdot,y)$ is lower semicontinous and convex and, for each $x \in N$, $g(x,\cdot)$ is upper semicontinous and concave. Then,*

$$\inf_{x\in M} \sup_{y\in N} g(x,y) = \sup_{y\in N} \inf_{x\in M} g(x,y).$$

We now apply Sion's minimax Theorem. First define

$$g((\pi,P),(\alpha,\beta)) = \int_{\Omega\times\Omega} c(x,y)\pi(\mathrm{d}x,\mathrm{d}y) + (\mathbb{E}_{P_*}[f(X)] - \mathbb{E}_P[f(X)])$$

$$+ \mathbb{E}_P[\alpha(X)] + \int_\Omega \beta(y)P_0(\mathrm{d}y) - \int_{\Omega\times\Omega} (\alpha(x) + \beta(y))\pi(\mathrm{d}x,\mathrm{d}y).$$

Next, for each $a \geq 1$, note that

$$R_0 = \inf_{(\pi,P)\in\mathcal{M}_+^a(\Omega\times\Omega)\times\mathcal{M}_+^a(\Omega)} \sup_{(\alpha,\beta,f)\in\mathcal{C}(\Omega)\times\mathcal{C}(\Omega)\times\mathcal{B}(\Omega)} \left\{ \int_{\Omega\times\Omega} c(x,y)\pi(\mathrm{d}x,\mathrm{d}y) + \right.$$

$$\left. (\mathbb{E}_{P_*}[f(X)] - \mathbb{E}_P[f(X)]) + \mathbb{E}_P[\alpha(X)] + \int_\Omega \beta(y)P_0(\mathrm{d}y) - \int_{\Omega\times\Omega} (\alpha(x) + \beta(y))\pi(\mathrm{d}x,\mathrm{d}y) \right\}.$$

Let spaces $M = \mathcal{M}_+^a(\Omega \times \Omega) \times \mathcal{M}_+^a(\Omega)$ and $N = \mathcal{C}(\Omega) \times \mathcal{C}(\Omega) \times \mathcal{B}(\Omega)$ be equipped with the product of weak topology and product of uniform topology, respectively. Both $M$ and $N$ are convex spaces and $M$ is compact by Prohorov's Theorem (see, e.g., [21, Theorem 2.4]), since $\Omega$ is compact.

Clearly, the function $g((\pi,P),(\alpha,\beta,f))$ is linear in $(\pi,P)$ and $(\alpha,\beta,f)$ so that $g(\cdot)$ is convex-concave as required by Sion's Theorem. We claim that $g(\cdot,(\alpha,\beta))$ is continous under the weak topology. For any $(\pi_n,P_n) \Rightarrow (\pi,P)$, we have for any continous and bounded functions $\phi_1$ and $\phi_2$

$$\int \phi_1\pi_n + \int \phi_2 P_n \to \int \phi_1\pi + \int \phi_2 P.$$

Since $c(\cdot), \alpha(\cdot), \beta(\cdot)$ and $g(\cdot)$ are continuous functions on a compact space, they are all bounded and therefore, by the definition of weak convergence, we immediately obtain that

$$g((\pi_n,P_n),(\alpha,\beta)) \to g((\pi,P),(\alpha,\beta)).$$

On the other hand, for any $(\alpha_n,\beta_n,f_n) \to (\alpha,\beta,f)$ uniformly, we have

$$g((\pi,P),(\alpha_n,\beta_n,f_n)) \to g((\pi,P),(\alpha,\beta,f)),$$

given that $\pi(\Omega \times \Omega) < \infty$ and $P(\Omega \times \Omega) < \infty$, by the bounded convergence theorem.

Hence, we now can apply Sion's duality:

$$
\begin{aligned}
R_0 \;=\; & \sup_{(\alpha,\beta,f)\in\mathcal{C}(\Omega)\times C(\Omega)\times \mathcal{LB}(\Omega)} \; \inf_{(\pi,P)\in\mathcal{M}_+^a(\Omega\times\Omega)\times\mathcal{M}_+^a(\Omega)} \left\{ \int_{\Omega\times\Omega} c(x,y)\pi(\mathrm{d}x,\mathrm{d}y)+ \right. \\
& \left( \mathbb{E}_{P_*}\left[f(X)\right] - \int_\Omega f(x)P(\mathrm{d}x) \right) + \int_\Omega \alpha(x)P(\mathrm{d}x) + \int_\Omega \beta(y)P_0(\mathrm{d}y) \\
& \left. - \int_{\Omega\times\Omega} \left(\alpha(x)+\beta(y)\right)\pi(\mathrm{d}x,\mathrm{d}y) \right\} \\
\;=\; & \sup_{(\alpha,\beta,f)\in\mathcal{C}(\Omega)\times C(\Omega)\times \mathcal{LB}(\Omega)} \; \inf_{(\pi,P)\in\mathcal{M}_+^a(\Omega\times\Omega)\times\mathcal{M}_+^a(\Omega)} \left\{ \left(\mathbb{E}_{\mathbb{P}_*}\left[f(X)\right] - \mathbb{E}_P\left[f(X)\right]\right) + \right. \\
& \left. \mathbb{E}_P\left[\alpha(X)\right] + \int_\Omega \beta(y)P_0(\mathrm{d}y) + \int_{\Omega\times\Omega} \left(c(x,y)-\alpha(x)-\beta(y)\right)\pi(\mathrm{d}x,\mathrm{d}y) \right\}.
\end{aligned}
$$

By choosing $\alpha=\beta=f=0$, we conclude that $R_0 \geq 0$.

We first claim that there is no incentive for the sup player to select functions $\alpha(\cdot)$ and $\beta(\cdot)$ such that $\alpha(x)+\beta(y) > c(x,y)$ for some $(x,y)\in\Omega\times\Omega$. In that case, let $\pi(\{x,y\})=a$, $\pi(\{x,y\}^c)=0$ and $P=P_*$. With $a\geq 1$ being chosen arbitrarily, this implies that

$$
R_0 \leq \int_\Omega f(x)P(\mathrm{d}x) + \int_\Omega \beta(y)P_0(\mathrm{d}y) - \left(\alpha(x)+\beta(y)-c(x,y)\right)a < 0.
$$

Therefore, we conclude that

$$
\begin{aligned}
R_0 \;=\; & \sup_{(\alpha,\beta,f)\in\mathcal{J}(c)\times \mathcal{LB}(\Omega)} \; \inf_{(\pi,P)\in\mathcal{M}_+^a(\Omega\times\Omega)\times\mathcal{M}_+^a(\Omega)} \left\{ \left( \mathbb{E}_{P_*}\left[f(X)\right] - \int_\Omega f(x)P(\mathrm{d}x) \right) + \right. \\
& \left. \int_\Omega \alpha(x)P(\mathrm{d}x) + \int_\Omega \beta(y)P_0(\mathrm{d}y) + \int_{\Omega\times\Omega} \left(c(x,y)-\alpha(x)-\beta(y)\right)\pi(\mathrm{d}x,\mathrm{d}y) \right\}.
\end{aligned}
$$

We then claim that $\alpha(x) < f(x)$ for some $x\in\Omega$ is impossible. By choosing $P(\{x\})=a$, we have $P(\{x\}^c)=0$, $\pi(\Omega\times\Omega)=0$ and $\beta(\cdot)=0$. By choosing sufficiently large $a$, we obtain

$$
R_0 \leq \mathbb{E}_{P_*}\left[f(X)\right] - a\left(f(x)-\alpha(x)\right) < 0.
$$

Therefore, we conclude that

$$
\begin{aligned}
R_0 \;=\; & \sup_{(\alpha,\beta,f)\in\mathcal{J}(c)\times \mathcal{LB}(\Omega), f\leq\alpha} \; \inf_{(\pi,P)\in\mathcal{M}_+^a(\Omega\times\Omega)\times\mathcal{M}_+^a(\Omega)} \left\{ \left( \mathbb{E}_{P_*}\left[f(X)\right] - \int_\Omega f(x)P(\mathrm{d}x) \right) + \right. \\
& \left\{ \left. \int_\Omega \alpha(x)P(\mathrm{d}x) + \int_\Omega \beta(y)P_0(\mathrm{d}y) + \int_{\Omega\times\Omega} \left(c(x,y)-\alpha(x)-\beta(y)\right)\pi(\mathrm{d}x,\mathrm{d}y) \right\}. \right.
\end{aligned}
$$

For the inner infimum, we can always choose $\pi(\Omega\times\Omega)=0$ and $P(\Omega)=0$. Notice that $\alpha^c(x)$ is a continuous function in the compact space and thus

$$
\begin{aligned}
R_0 \;\leq\; & \sup_{(\alpha,\beta,f)\in\mathcal{J}(c)\times\mathcal{LB}(\Omega), f\leq\alpha} \; \mathbb{E}_{P_*}\left[f(X)\right] + \int_\Omega \beta(y)P_0(\mathrm{d}y) \\
\;=\; & \sup_{(\alpha,\beta,f)\in\mathcal{J}(c)\times\mathcal{LB}(\Omega), f\leq\alpha} \; \mathbb{E}_{P_*}\left[f(X)\right] - \int_\Omega \alpha^c(y)P_0(\mathrm{d}y) \\
\;\leq\; & \sup_{f\in\mathcal{LB}(\Omega)} \mathbb{E}_{P_*}\left[f(X)\right] - \int_\Omega f^c(y)P_0(\mathrm{d}y),
\end{aligned}
$$

where the last inequality follows from

$$
\sup_{x'\in\Omega}\left\{\alpha(x')-c(x',x)\right\} \geq \sup_{x'\in\Omega}\left\{f(x')-c(x',x)\right\}, \quad \text{if } \alpha(\cdot)\geq f(\cdot).
$$

## Appendix A.2  Proof of Proposition 1

By the Kantorovich-Rubinstein duality [24, Theorem 5.10], we have

$$\max_{\theta \in \Theta} \mathcal{W}_2 \left( \theta_\sharp P, \theta_\sharp Q \right)^2$$

$$= \max_{\theta \in \Theta} \max_{f \in \mathcal{C}_b(\mathbb{R})} \left( \mathbb{E}_P \left[ f \left( \theta^\top X \right) \right] - \mathbb{E}_Q \left[ \sup_{t \in \mathbb{R}} f(t) - \left( \theta^\top X - t \right)^2 \right] \right)$$

$$= \max_{\theta \in \Theta, f \in \mathcal{C}_b(\mathbb{R})} \left( \mathbb{E}_P \left[ f \left( \theta^\top X \right) \right] - \mathbb{E}_Q \left[ \sup_{y \in \mathbb{R}^d} f(\theta^\top y) - \left( \theta^\top X - \theta^\top y \right)^2 \right] \right).$$

On the other hand, for any $\theta$ with $\|\theta\|_2 = 1$ and $f \in \mathcal{C}_b(\mathbb{R})$, we obtain

$$\sup_{y \in \mathbb{R}^d} \left( f(\theta^\top y) - \|x - y\|_2^2 \right) = \sup_{y \in \mathbb{R}^d} \left( f(\theta^\top y) - \inf_{y' : \theta^\top y' = \theta^\top y} \|x - y'\|_2^2 \right)$$

$$= \sup_{y \in \mathbb{R}^d} \left( f(\theta^\top y) - \left( \theta^\top x - \theta^\top y \right)^2 \right).$$

Therefore, we have

$$\max_{\theta \in \Theta} \mathcal{W}_2 \left( \theta_\sharp P, \theta_\sharp Q \right)^2 = \sup_{f \in \mathcal{B}_{\max}(\mathbb{R}^d, \Theta)} \left( \mathbb{E}_P \left[ f(X) \right] - \mathbb{E}_Q \left[ \sup_{y \in \mathbb{R}^d} \left( f(y) - \|x - y\|_2^2 \right) \right] \right),$$

which completes the proof.

## Appendix A.3  Proof of Theorem 2

Define

$$\mathcal{UB}_i(\Omega) = \left\{ f(\theta_i^\top \cdot) \in \mathcal{LB}_i(\Omega) : \mathbb{E}_{P_*} \left[ f'(\theta_i^\top X)^2 \right] = 1, f(0) = 0 \right\},$$

and accordingly

$$\mathcal{UB}(\Omega) = \left\{ f(\cdot) \in \mathcal{LB}(\Omega) : \mathbb{E}_{P_*} \left[ \|\nabla_X f(X)\|_2^2 \right] = 1, f(0) = 0 \right\}.$$

Since $P_*$ has a non-zero density, we have $\mathcal{LB}(\Omega) = \left\{ \lambda f(\cdot) + c \,|\, \lambda, c \in \mathbb{R}, f(\cdot) \in \mathcal{UB}(\Omega) \right\}.$

By Theorem 1, we obtain

$$R_n = \sup_{\lambda \in \mathbb{R}} \sup_{f \in \mathcal{UB}(\Omega)} \left\{ E_P \left[ \lambda f(X) \right] - \frac{1}{n} \sum_{i=1}^n \left[ \sup_{\Delta + X_i \in \Omega} \left\{ \lambda f(X_i + \Delta) - \|\Delta\|_2^2 \right\} \right] \right\}$$

$$= \sup_{\lambda \in \mathbb{R}} \sup_{f \in \mathcal{UB}(\Omega)} \left\{ E_P \left[ \lambda f(X) \right] - \frac{1}{n} \sum_{i=1}^n \lambda f(X_i) \right.$$

$$\left. - \frac{1}{n} \sum_{i=1}^n \left[ \sup_{\Delta + X_i \in \Omega} \left\{ \lambda f(X_i + \Delta) - \lambda f(X_i) - \|\Delta\|_2^2 \right\} \right] \right\}.$$

Let $H_n^f = n^{-1/2} \left( \sum_{i=1}^n f(X_i) - E_P \left[ f(X) \right] \right)$. By rescaling the variables $\lambda$ and $\Delta$, we have
$n R_n =$

$$\sup_{\lambda \in \mathbb{R}} \sup_{f \in \mathcal{UB}(\Omega)} \left\{ -2\lambda H_n^f - \frac{1}{n} \sum_{i=1}^n \left[ \sup_{X_i + \Delta/\sqrt{n} \in \Omega} \left\{ 2\lambda \sqrt{n} \left( f(X_i + \Delta/\sqrt{n}) - f(X_i) \right) - \|\Delta\|_2^2 \right\} \right] \right\}.$$

The following sequence of results will then be useful in proving Theorem 2. To simplify the notation, we denote

$$M_n(\lambda, f) = \frac{1}{n} \sum_{i=1}^n \left[ \sup_{X_i + \Delta/\sqrt{n} \in \Omega} \left\{ 2\lambda \sqrt{n} \left( f(X_i + \Delta/\sqrt{n}) - f(X_i) \right) - \|\Delta\|_2^2 \right\} \right].$$

Henceforth, we refer to a function class as a Donsker class or a Glivenko-Cantelli class if the function class is a $P$-Donsker class or $P$-Glivenko-Cantelli class for all Borel measure $P$ supported on the sample domain $\Omega$.

**Proposition A1.** *There exists $M_B < \infty$ such that*

$$\mathcal{UB}(\Omega) \subset \mathcal{F}^{M_B} := \left\{ \sum_{i=1}^{K} \xi_i f_i \left( \theta_i^\top \cdot \right) : |\xi_i| \leq M_B \text{ and } f_i \left( \theta_i^\top \cdot \right) \in \mathcal{UB}_i(\Omega) \right\},$$

*and thus $\mathcal{UB}(\Omega)$ is a Donsker class.*

**Proposition A2.** *The function class $\mathcal{UB}'(\Omega) = \left\{ \|\nabla_x f(\cdot)\|_2^2 : f \in \mathcal{UB}(\Omega) \right\}$ is a Glivenko-Cantelli class. Furthermore, $\mathcal{UB}'_\epsilon(\Omega) = \left\{ \|\nabla_x f(\cdot)\|_2^2 \, \mathbb{I}\{B_\epsilon(\cdot) \subset \Omega\} : f \in \mathcal{UB}(\Omega) \right\}$ are also Glivenko-Cantelli classes, where $B_\epsilon(x)$ is a closed ball around $x$ with radius $\epsilon$, i.e., $B_\epsilon(x) = \{y \in \mathbb{R}^d : \|x - y\| \leq \epsilon\}$.*

**Proposition A3.** *For every $\epsilon > 0$, there exists $n_0 > 0$ and $b \in (0, \infty)$ such that*

$$\mathbb{P}\left( \sup_{|\lambda| > b} \sup_{f \in \mathcal{UB}(\Omega)} \left\{ -2\lambda H_n^f - M_n(\lambda, f) \right\} > 0 \right) \leq \epsilon,$$

*for all $n \geq n_0$.*

**Proposition A4.** *We have*

$$\sup_{|\lambda| \leq b} \sup_{f \in \mathcal{UB}(\Omega)} \left( \left| M_n(\lambda, f) - \lambda^2 \right| \right) \to 0,$$

*almost surely.*

Based on Propositions A1 – A4, we are now ready to present the proof of Theorem 2.

*Proof of Theorem 2.* By the uniform convergence property of Donsker classes and Glivenko-Cantelli classes, we have

$$\sup_{|\lambda| \leq b} \sup_{f \in \mathcal{UB}(\Omega)} \left\{ -2\lambda H_n^f - M_n(\lambda, f) \right\} \Rightarrow \sup_{|\lambda| \leq b} \sup_{f \in \mathcal{UB}(\Omega)} \left\{ -2\lambda H^f - \lambda^2 \right\},$$

where $H^f$ is a Gaussian process with $H^f \sim \mathcal{N}(0, \text{var}(f(X)))$ and $\text{cov}(H^{f_1}, H^{f_2}) = \text{cov}(f_1(X), f_2(X))$. Furthermore, Proposition A3 implies

$$\mathbb{P}\left( \sup_{\lambda \in \mathbb{R}} \sup_{f \in \mathcal{UB}(\Omega)} \left\{ -2\lambda H_n^f - M_n(\lambda, f) \right\} = \sup_{|\lambda| \leq b} \sup_{f \in \mathcal{UB}(\Omega)} \left\{ -2\lambda H_n^f - M_n(\lambda, f) \right\} \right) \geq 1 - \epsilon.$$

Therefore, by Slutsky's Theorem, we obtain

$$\sup_{\lambda \in \mathbb{R}} \sup_{f \in \mathcal{UB}(\Omega)} \left\{ -2\lambda H_n^f - M_n(\lambda, f) \right\} \Rightarrow \sup_{\lambda \in \mathbb{R}} \sup_{f \in \mathcal{UB}(\Omega)} \left\{ -2\lambda H^f - \lambda^2 \right\}$$

$$= \sup_{f \in \mathcal{LB}(\Omega)} \left\{ -2H^f - \mathbb{E}_{P_*} \left[ \|\nabla_X f(X)\|_2^2 \right] \right\}.$$

$\square$

### Appendix A.3.1    Proofs of Propositions A1 – A4

**Lemma A1.** *$\mathcal{UB}_i(\Omega)$ is a Donsker class.*

*Proof.* By Assumptions 2 and 3, we have for any $f(\theta_i^\top \cdot) \in \mathcal{UB}_i(\Omega)$,

$$\sup_{x \in \theta_i^\top \Omega} |f''(x)| \leq M \sqrt{\int_\Omega f'(\theta_i^\top x)^2 \mathrm{d}x} \leq \frac{M}{\sqrt{\underline{b}}} \sqrt{\mathbb{E}_{P_*} \left[ f'(\theta_i^\top X)^2 \right]} = \underline{b}^{-1/2} M,$$

where $\theta_i^\top \Omega$ is defined as $\theta_i^\top \Omega = \{\theta_i^\top x : x \in \Omega\}$. Notice that $|f'(x)| = \left| f'(0) + \int_0^x f''(s)\mathrm{d}s \right|$ and

$$|f'(0)| + M |x| \geq |f'(x)| \geq |f'(0)| - \underline{b}^{-1/2} M |x|. \tag{A.2}$$

Recall that $\mathbb{E}_{P_*}\left[f'(\theta_i^\top X)^2\right] = 1$, and thus

$$1 = \mathbb{E}_{P_*}\left[f'(\theta_i^\top X)^2\right] \geq \mathbb{E}_{P_*}\left[\left(|f'(0)| - \underline{b}^{-1/2}M\left|\theta_i^\top X\right|\right)^2\right].$$

Since $\mathbb{E}_{P_*}\left[f'\left(\theta_i^\top X\right)^2\right]$ is bounded, we have $|f'(0)|$ is bounded from above. Furthermore, by (A.2), we conclude that $|f'(x)|$ is bounded from above in $\theta_i^\top\Omega$, which means $f(x)$ is a Lipschitz function with a bounded Lipschitz constant. By Example 19.9 in [21], we have the space of one-dimensional bounded Lipschitz functions with $f(0) = 0$ is a Donsker class for all Borel probability measures supported in $\theta_i^\top\Omega$. Recalling $(\theta_i)_\sharp P$ is the push-forward measure from $\mathcal{P}(\Omega)$ to $\mathcal{P}\left(\theta_i^\top\Omega\right)$ defined in (4), we then obtain $\mathbb{E}_{(\theta_i)_\sharp P}\left[f(X)\right] = \mathbb{E}_P\left[f\left(\theta_i^\top X\right)\right]$. We conclude $\mathcal{UB}_i(\Omega)$ is also a Donsker class for all Borel probability measures supported in $\Omega$. $\square$

**Lemma A2.** *For two positive semidefinite matrices $A, B \in \mathbb{R}_{K \times K}$, define the Hadamard product $A \circ B$ as*

$$(A \circ B)_{ij} = A_{ij}B_{ij}.$$

*Then, if $B_{ii} = 1, \forall i$, we have $\sigma_K(A \circ B) \geq \sigma_K(A)$, where $\sigma_K(\cdot)$ denotes the smallest eigenvalue of the matrix.*

*Proof.* Denote the $k$-th leading principle minor of a matrix $A$ as $[A]_{1:k,1:k}$. Let $\mu = \sigma_K(A)$. Then, $A - \mu I$ is positive semidefinite. By Oppenheim's inequality (see, e.g., [14, Theorem 7.8.16]), we have $\det\left([A - \mu I]_{1:k,1:k} \circ [B]_{1:k,1:k}\right) \geq \det[A - \mu I]_{1:k,1:k} \geq 0$. Hence, $(A - \mu I) \circ B = A \circ B - \mu I$ is also positive semi-definite and $\sigma_K(A \circ B) \geq \mu$. $\square$

We are now ready to present the proof of Proposition A1.

*Proof of Proposition A1.*

$$\mathbb{E}_{P_*}\left[\|\nabla_x f(x)\|_2^2\right] = \mathbb{E}_{P_*}\left[\left\|\sum_{i=1}^K \lambda_i f_i'(\theta_i^\top X)\theta_i\right\|_2^2\right] = \xi^T(A \circ B)\xi,$$

where

$$B_{ij} = \mathbb{E}\left[f_i'(\theta_i^\top X)f_j'(\theta_j^\top X)\right] \text{ and } A_{ij} = \theta_i^\top\theta_j.$$

Since, by construction, $A$ and $B$ are both positive semidefinite matrices, then by Lemma A2 we have

$$1 = \xi^T(A \circ B)\xi \geq \left(\xi^\top\xi\right)\sigma_K(A).$$

Since $\theta_1, \ldots, \theta_K$ are linearly independent, $\sigma_K(A) > 0$. Letting $M_B = \sqrt{1/\sigma_K(A)}$, we then have $|\xi_i| \leq \sqrt{1/\sigma_K(A)}$ and $\mathcal{F}^{M_B}$ is a Donsker class by Theorem 2.10.3 in [22]. Therefore, by Theorem 2.10.1 in [22], we have $\mathcal{UB}(\Omega)$ is a Donsker class. $\square$

**Lemma A3.** *For any function $f \in \mathcal{UB}(\Omega)$, define the Hessian matrix of $H^f(x)$ as $H_{ij}^f(x) = \frac{\partial}{\partial x_i \partial x_j}f(x)$. Then, there exist universal constants $M_1$ and $M_2$ such that $\sup_{f \in \mathcal{UB}(\Omega)}\sup_{x \in \Omega}\|\nabla_x f(x)\|_2 \leq M_1$ and $\sup_{f \in \mathcal{UB}(\Omega)}\sup_{x \in \Omega}\left\|H^f(x)\right\|_F \leq M_2$.*

*Proof.* For $f(x) = \sum_{i=1}^K \lambda_i f_i(\theta_i^\top x)$, we have

$$\|\nabla_x f(x)\|_2 \leq \sum_{i=1}^K \left\|\lambda_i f_i'(\theta_i^\top x)\theta_i\right\|_2 \leq KM_B \sup_{1 \leq i \leq K}\sup_{x \in \theta_i^\top\Omega}|f_i'(x)|,$$

where, by Lemma A1, $\sup_{1 \leq i \leq K}\sup_{x \in \theta_i^\top\Omega}|f_i'(x)| < \infty$. Furthermore, we obtain

$$\left|H_{ij}^f(x)\right| \leq \sum_{k=1}^K \left|\lambda_k f_k''(\theta_k^\top x)\theta_k^{(i)}\theta_k^{(j)}\right| \leq \underline{b}^{-1/2}MKM_B.$$

Let $M_1 = KM_B\sup_{1 \leq i \leq K}\sup_{x \in \theta_i^\top\Omega}|f_i'(x)|$, $M_2 = d^2\underline{b}^{-1/2}MKM_B$, and thus $\sup_f\sup_x\|\nabla_x f(x)\|_2 \leq M_1$ and $\left\|H^f(x)\right\|_F \leq M_2$ for any $f \in \mathcal{UB}(\Omega)$ and $x \in \Omega$. $\square$

We are now ready to present the proof of Proposition A2.

*Proof of Proposition A2.* Consider

$$
\begin{aligned}
\frac{\partial \|\nabla_x f(x)\|_2^2}{\partial x_k} &= \frac{\partial \left\|\sum_{i=1}^K \lambda_i f_i'(\theta_i^\top x)\theta_i\right\|_2^2}{\partial x_k} \\
&= \sum_{i=1}^K \sum_{j=1}^K \lambda_i \lambda_j \theta_i^\top \theta_j \left( f_i''(\theta_i^\top x) f_j'\left(\theta_j^\top x\right)\theta_i^{(k)} + f_i'(\theta_i^\top x) f_j''\left(\theta_j^\top x\right)\theta_j^{(k)}\right) \\
&\leq 2K^2 M_B^2 \left( \sup_{1 \leq i \leq K} \sup_{x \in \theta_i^\top \Omega} |f_i'(x)| \right) \underline{b}^{-1} M.
\end{aligned}
$$

Therefore, the partial derivative $\frac{\partial \|\nabla_x f(x)\|_2^2}{\partial x_k}$ is bounded. By Example 19.9 and Theorem 19.4 in [21], we have $\mathcal{UB}'(\Omega)$ is a Glivenko-Cantelli class. Furthermore, since the bracketing number of $\mathcal{UB}'_\epsilon(\Omega)$ is bounded by that of $\mathcal{UB}'(\Omega)$, we have $\mathcal{UB}'_\epsilon(\Omega)$ are also Glivenko-Cantelli classes. $\qquad\square$

We are now ready to present the proof of Proposition A3.

*Proof of Proposition A3.* By Taylor expansion, we have

$$
\left| f(X_i + \Delta/\sqrt{n}) - f(X_i) - \frac{1}{\sqrt{n}}\left(\nabla_X f(X_i)\right)^\top \Delta \right| \leq M_2 \|\Delta\|_2^2 /n.
$$

By substituting $\Delta_i = c_1 \nabla_X f(X_i)$, we obtain

$$
\begin{aligned}
&\sup_{X_i + \Delta/\sqrt{n} \in \Omega} \left\{ 2\lambda\sqrt{n}\left(f(X_i + \Delta/\sqrt{n}) - f(X_i)\right) - \|\Delta\|_2^2 \right\} \\
&\geq \left( 2|\lambda|\left( (\nabla_X f(X_i))^\top \Delta_i - M_2 \|\Delta_i\|_2^2/\sqrt{n} \right) - \|\Delta_i\|_2^2 \right) \mathbb{I}\{X_i + \Delta_i/\sqrt{n} \in \Omega\} \\
&= \left( 2|\lambda|c_1 - c_1^2 - 2|\lambda|c_1^2 M_2/\sqrt{n} \right) \|\nabla_X f(X_i)\|_2^2 \, \mathbb{I}\{X_i + c_1 \nabla_X f(X_i)/\sqrt{n} \in \Omega\} \\
&\geq \left( 2|\lambda|c_1 - c_1^2 - 2|\lambda|c_1^2 M_2/\sqrt{n} \right) \|\nabla_X f(X_i)\|_2^2 \, \mathbb{I}\{B_{c_1 M_1/\sqrt{n}}(X_i) \in \Omega\}.
\end{aligned}
$$

We then derive

$$
\begin{aligned}
&\sup_{|\lambda|>b} \sup_{f \in \mathcal{UB}(\Omega)} \left\{ -2\lambda H_n^f - M_n(\lambda, f) \right\} \\
&\leq \sup_{|\lambda|>b} \sup_{f \in \mathcal{UB}(\Omega)} \Big\{ -2\lambda H_n^f \\
&\quad - \frac{1}{n}\sum_{i=1}^n \left(2|\lambda|c_1 - c_1^2 - 2|\lambda|c_1^2 M_2/\sqrt{n}\right) \|\nabla_X f(X_i)\|_2^2 \, \mathbb{I}\{B_{c_1 M_1/\sqrt{n}}(X_i) \in \Omega\} \Big\} \\
&\leq \sup_{|\lambda|>b} \sup_{f \in \mathcal{UB}(\Omega)} \Big\{ -2\lambda H_n^f \\
&\quad - |\lambda| \frac{1}{n}\sum_{i=1}^n \left(2c_1 - \frac{c_1^2}{b} - 2c_1^2 M_2/\sqrt{n}\right) \|\nabla_X f(X_i)\|_2^2 \, \mathbb{I}\{B_{c_1 M_1/\sqrt{n}}(X_i) \in \Omega\} \Big\} \\
&\leq \sup_{|\lambda|>b} |\lambda| \Bigg( 2 \sup_{f \in \mathcal{UB}(\Omega)} \left|H_n^f\right| \\
&\quad - \left(2c_1 - \frac{c_1^2}{b} - 2c_1^2 M_2/\sqrt{n}\right) \inf_{f \in \mathcal{UB}(\Omega)} \left( \frac{1}{n}\sum_{i=1}^n \|\nabla_X f(X_i)\|_2^2 \right) \mathbb{I}\{B_{c_1 M_1/\sqrt{n}}(X_i) \in \Omega\} \Bigg),
\end{aligned}
$$

where $B_\varepsilon(X_i) = \{y \in \mathbb{R}^d : \|y - X_i\|_2 \leq \varepsilon\}$. Since $\mathcal{UB}(\Omega)$ is a Donsker class, we have

$$
\sup_{f \in \mathcal{UB}(\Omega)} \left|H_n^f\right| \Rightarrow \sup_{f \in \mathcal{UB}(\Omega)} \left|H^f\right|,
$$

where $\sup_{f\in\mathcal{UB}(\Omega)}\left|H^f\right| < \infty$ almost surely. Hence, there exist $n_1$ and $b'$ such that, for $n \geq n_1$,
$\mathbb{P}\left(\sup_{f\in\mathcal{UB}(\Omega)}\left|H_n^f\right| > b'\right) < \epsilon/2$.

Since $P_*(\Omega^\circ) = 1$, we can choose $\varepsilon' > 0$ such that

$$\mathbb{E}_{P_*}\left[\|\nabla_X f(X_i)\|_2^2 \, \mathbb{I}\{B_{\varepsilon'}(X_i) \in \Omega\}\right] > \frac{3}{4}.$$

By Lemma A1, there exists $n_2 > n_1$ such that, for $n \geq n_2$,

$$\mathbb{P}\left(\inf_{f\in\mathcal{UB}(\Omega)}\left(\frac{1}{n}\sum_{i=1}^{n}\|\nabla_X f(X_i)\|_2^2\right)\mathbb{I}\{B_{\varepsilon'}(X_i) \in \Omega\} \leq 1/2\right) < \epsilon/2.$$

Letting $c_1 = 4b'$, $b = 2c_1$ and for $n > n_0 = \max\{(4c_1 M_2)^2, (c_1 M_1/\varepsilon')^2, n_2\}$, we then have

$$\mathbb{P}\left(\sup_{|\lambda|>b}\sup_{f\in\mathcal{UB}(\Omega)}\left\{-2\lambda H_n^f - M_n(\lambda, f)\right\} > 0\right)$$

$$\leq \quad \mathbb{P}\left(\inf_{f\in\mathcal{UB}(\Omega)}\left(\frac{1}{n}\sum_{i=1}^{n}\|\nabla_X f(X_i)\|_2^2\right)\mathbb{I}\{B_{\varepsilon'}(X_i) \in \Omega\} \leq 1/2\right)$$

$$+ \quad \mathbb{P}\left(\sup_{f\in\mathcal{UB}(\Omega)}\left|H_n^f\right| > b'\right)$$

$$< \quad \epsilon.$$

$\square$

We are now ready to present the proof of Proposition A4.

*Proof of Proposition A4.* First note that, for $|\lambda| \leq b$, we have

$$2\lambda\sqrt{n}\left(f(X_i + \Delta/\sqrt{n}) - f(X_i)\right) - \|\Delta\|_2^2$$

$$= \quad 2\lambda\int_0^1\left(\nabla_X f(X_i + n^{-1/2}\Delta u)\right)^\top \Delta \mathrm{d}u - \|\Delta\|_2^2$$

$$\leq \quad 2b\int_0^1\left\|\nabla_X f(X_i + n^{-1/2}\Delta u)\right\|_2 \|\Delta\|_2 \, \mathrm{d}u - \|\Delta\|_2^2$$

$$\leq \quad 2bM_1\|\Delta\|_2 - \|\Delta\|_2^2.$$

Therefore, we only need to consider $\|\Delta\|_2 \leq 2bM_1$. Recalling the Taylor expansion

$$\sup_{\|\Delta\|\leq 2bM_1}\left|f(X_i + \Delta/\sqrt{n}) - f(X_i) - n^{-1/2}\left(\nabla_X f(X_i)\right)^\top \Delta\right| \leq \frac{M_2}{n}\|\Delta\|_2^2 < \frac{M_2}{n}(2bM_1)^2,$$

we then obtain

$$\sup_{|\lambda|\leq b}\sup_{f\in\mathcal{UB}(\Omega)}\left(\sup_{X_i+\Delta/\sqrt{n}\in\Omega, \|\Delta\|\leq 2bM_1} M_n(\lambda, f)\right.$$

$$\left.-\frac{1}{n}\sum_{i=1}^{n}\sup_{X_i+\Delta/\sqrt{n}\in\Omega, \|\Delta\|\leq 2bM_1}\left(2\lambda\left(\nabla_X f(X_i)\right)^\top \Delta - \|\Delta\|_2^2\right)\right)$$

$$\leq \quad M_2(2bM_1)^2/\sqrt{n} \to 0.$$

Furthermore, we have

$$\lambda^2\|\nabla_X f(X_i)\|_2^2\,\mathbb{I}\{B_{\lambda M_1/\sqrt{n}}(X_i) \in \Omega\}$$

$$\leq \quad \sup_{X_i+\Delta/\sqrt{n}\in\Omega, \|\Delta\|\leq 2bM_1}\left(2\lambda\left(\nabla_X f(X_i)\right)^\top \Delta - \|\Delta\|_2^2\right) \leq \lambda^2\|\nabla_X f(X_i)\|_2^2.$$

Since $P_*(\Omega^\circ) = 1$, we obtain

$$\frac{1}{n}\sum_{i=1}^{n}\left(\lambda^2 \|\nabla_x f(X_i)\|_2^2 \, \mathbb{I}\{B_{\lambda M_1/\sqrt{n}}(X_i) \in \Omega\} - \lambda^2 \|\nabla_x f(X_i)\|_2^2\right) \to 0,$$

almost surely. Therefore, we conclude

$$\sup_{|\lambda| \leq b} \sup_{f \in \mathcal{UB}(\Omega)} \left| M_n(\lambda, f) - \frac{1}{n}\sum_{i=1}^{n} \lambda^2 \|\nabla_x f(X_i)\|_2^2 \right| \to 0,$$

almost surely. By Lemma A2, we have the desired results. $\qquad\square$

## Appendix A.4    Proof of Theorem 3

We apply a line of arguments similar to steps 2 and 3 in the proof of Theorem 1.3 in [23]. To simplify the notation, we define $\mathcal{V} := R_n$ and

$$\mathcal{D} := \sup_{f \in \mathcal{LB}(\mathbb{R}^d)} \{\mathbb{E}_{P_*}[f(X)] - \mathbb{E}_{P_n}[f^c(X)]\}.$$

Since $\mathcal{V} \geq \mathcal{D}$ is proved in Appendix A.1.1, we only need to show $\mathcal{D} \geq \mathcal{V}$. The strategy of this proof is to pick a series of large compact sets, so that we can approximate the solution to the primal problem by restricting the functions $c(\cdot, \cdot)$ and $f$ on the compact set.

We next apply strong duality for the compact problem and then show that the dual optimal value $\mathcal{D}$ can be approximated by the dual optimal value of the compact problem, when we apply the truncation to the cost function $c_a(x, y) = \min\{a, c(x, y)\}$. Finally, we show that the optimal value with the cost function $c_a(x, y)$ converges to the optimal value with the cost function $c(x, y)$.

### Appendix A.4.1    Primal Approximation

For any $\epsilon > 0$, we pick a large compact set $\mathcal{K}$ such that

$$P_*(\mathcal{K}) > 1 - \epsilon, \ P_n(\mathcal{K}) = 1 \qquad \text{and} \qquad \frac{1}{n}\sum_{i=1}^{n} \mathbb{E}_{P_*}[c(X_i, X)\mathbb{I}\{X \in \mathcal{K}^c\}] < \epsilon.$$

Define the measure $P_\mathcal{K}$ supported on $\mathcal{K}$ with $P_\mathcal{K}(A) = P_*(A)/P_*(\mathcal{K})$ for any Borel measurable set $A \subset \mathcal{K}$. Then, consider the primal problem restricted in space $\mathcal{K}$:

$$\mathcal{V}_\mathcal{K} = \inf_{P \in \mathcal{P}(\mathcal{K})} \left\{\mathcal{D}_c(P_n, P) : \mathbb{E}_{P_*}[f(X)] = \mathbb{E}_{P_\mathcal{K}}[f(X)] \text{ for all } f \in \mathcal{B}(\mathbb{R}^d)\big|_\mathcal{K}\right\}, \qquad (\text{A.3})$$

where $\mathcal{B}(\mathcal{X})\big|_\mathcal{K}$ is the restriction of $\mathcal{B}(\mathbb{R}^d)$ on set $\mathcal{K}$. Notice that, for any feasible solution $P \in \mathcal{P}(\mathcal{K})$ of problem (A.3), we can construct

$$P'(A) = P(A \cap \mathcal{K}) P_*(\mathcal{K}) + P_*(A \cap \mathcal{K}^c),$$

which is a feasible solution of problem (1). Let $\pi_\mathcal{K}$ be the coupling between $P_n$ and $P$. Then, we can define a coupling between $P_n$ and $P'$ as $\pi^\epsilon$:

$$\pi^\epsilon(\{X_i\}, A) = \pi_\mathcal{K}(\{X_i\}, A \cap \mathcal{K}) P_*(\mathcal{K}) + \frac{1}{n}P_*(A \cap \mathcal{K}^c)$$

for $i = 1, 2, \ldots, n$ and any Borel measurable set $A \subset \mathcal{X}$. Then, for every feasible $P$, we have

$$\mathcal{V} \leq \mathcal{D}_c(P_n, P') \leq P_*(\mathcal{K}) \mathcal{D}_c(P_n, P) + \frac{1}{n}\sum_{i=1}^{n} \mathbb{E}_{P_*}[c(X_i, X)\mathbb{I}\{X \in \mathcal{K}^c\}].$$

Therefore, we obtain $\mathcal{V} \leq \mathcal{V}_\mathcal{K} + \epsilon$.

## Appendix A.4.2  Dual Approximation

We first find unit vectors $\theta_{K+1}, \theta_{K+2}, \ldots, \theta_d$ such that $\theta_1, \theta_2, \ldots, \theta_d$ are linearly independent and thus are a basis of $\mathbb{R}^d$. Define compact sets $\mathcal{K}_m$ as

$$\mathcal{K}_m = \bigcap_{i=1}^{d} \left\{ x \in \mathbb{R}^d \mid \theta_i^\top x \in [-m, m] \right\}.$$

It is easy to see that $\mathcal{K}_m$ is a nonempty compact set for $m > 0$, given $\theta_1, \theta_2, \ldots, \theta_d$ are linearly independent. We pick $m$ sufficiently large such that $P_n(\mathcal{K}_m) = 1$. Define the dual problem

$$\mathcal{D}_m = \sup_{f \in \mathcal{LB}(\mathbb{R}^d)|_{\mathcal{K}_m}} \left\{ \mathbb{E}_{P_{\mathcal{K}_m}} [f(X)] - \mathbb{E}_{P_n} [f^c(X)] \right\}. \tag{A.4}$$

Then, for any $f(x) = \sum_{i=1}^{K} f_i\left(\theta_i^\top x\right) \in \mathcal{LB}(\mathbb{R}^d)\big|_{\mathcal{K}_m}$, we define $\bar{f}(x)$ as the extension of $f(x)$ to $\mathbb{R}^d$: for any $z \in \mathbb{R}^d$, let $x^*$ be the unique solution of the linear equation system

$$\theta_i^\top x = \max\left\{\min\left\{\left(\theta_i^\top z\right), m\right\}, -m\right\}, \; i = 1, 2, \ldots, d;$$

then, $x^* \in \mathcal{K}_m$ and let $\bar{f}(z) = f(x^*) = \sum_{i=1}^{K} f_i\left(\theta_i^\top x^*\right)$. Therefore, $\bar{f}(z) \in \mathcal{LB}(\mathbb{R}^d)$.

We consider the truncated cost function $c_a(\cdot) = \min\{a, c(\cdot)\}$ for $0 < a < \infty$. Let $f$ be an $\epsilon$-optimizer of problem (A.4) with the cost function $c_a(\cdot)$. Since $\mathcal{D}_m \geq 0$, there exist $x_0 \in \mathcal{K}_m$ and $y_0 \in \{X_i\}_{i=1}^{n}$ such that (assuming $0 < \epsilon < 1$)

$$f(x_0) - f^{c_a}(y_0) \geq -1.$$

Without loss of generality, we assume $f_i(\theta_i^\top x_0) \geq -1/K$ for $i = 1, 2, \ldots, K$ and $f^c(y_0) \leq 1$. Then, we obtain

$$
\begin{aligned}
f(x) &\leq f^{c_a}(y_0) + c_a(x, x_0) \leq a + 1, \text{ for } x \in \mathcal{K}_m, \\
f^{c_a}(x) &\geq f(x_0) - c_a(x, x_0) \geq -a - 1, \text{ and} \\
f^{c_a}(x) &= \sup_{y \in \mathcal{K}_m} f(y) - c_a(x, y) \leq a + 1, \text{ for } x \in \{X_i\}_{i=1}^{n}.
\end{aligned}
$$

By construction, we have $\bar{f}(x) \leq a + 1$ for any $x \in \mathbb{R}^d$. Since $\left\{ x \in \mathbb{R}^d : c(x, x_0) \leq a \right\}$ is compact, we are able to pick a sufficiently large $\mathcal{K}_m$ such that

$$\inf_{x \in \mathcal{K}_m^c, y \in \{X_i\}_{i=1}^{n}} c_a(x, y) = a.$$

Therefore, we obtain $\bar{f}^{c_a}(x) = f^{c_a}(x)$ for $x \in \{X_i\}_{i=1}^{n}$.

Then, for $z \in \mathbb{R}$ and any $j = 1, 2, \ldots, K$, let $x'$ be the unique solution of the linear system

$$
\begin{aligned}
\theta_j^\top x &= z; \\
\theta_i^\top x &= \theta_i^\top x_0, \; i = 1, 2, \ldots j-1, j+1, \ldots, d.
\end{aligned}
$$

Since $\sum_{i=1}^{K} \bar{f}_i(\theta_i^\top x') = \bar{f}(x') \leq a + 1$, we have

$$\bar{f}_j(z) \leq a + 1 - \sum_{i=1, i\neq j}^{K} \bar{f}_i(\theta_i^\top x') \leq a + 2. \tag{A.5}$$

Furthermore, we claim $g(x) = \sum_{i=1}^{K} \max\left\{ \bar{f}_i(\theta_i^\top x), -K(a+2) \right\}$ is a valid $\epsilon$-optimizer with $g^{c_a}(x) = \bar{f}^{c_a}(x)$ for $x \in \{X_i\}_{i=1}^{n}$. This is because for any $y_0 \in \{X_i\}_{i=1}^{n}$, if $\bar{f}_i\left(\theta_i^\top x\right) \leq -K(a+2)$, we have

$$\bar{f}(x) - c(x, y_0) \leq -K(a+2) + (K-1)(a+2) = -(a+2) < \bar{f}^{c_a}(y_0).$$

Therefore, $\mathbb{E}_{P_{\mathcal{K}_m}} [g(x)|_{\mathcal{K}_m}] \geq \mathbb{E}_{P_{\mathcal{K}_m}} [f(x)]$ with bounds $a + 1 \geq g(x) \geq -K^2(a+2)$. Finally, by picking sufficiently large $\mathcal{K}_m$ with $P_*(\mathcal{K}_m) > 1 - \epsilon$, we obtain

$$
\begin{aligned}
& \mathbb{E}_{P_*} [g(X)] - \mathbb{E}_{P_n} [g^{c_a}(X)] \\
\geq \; & (1 - \epsilon) \mathbb{E}_{P_{\mathcal{K}_m}} [f(X)] + \mathbb{E}_{P_*} [g(X)\mathbb{I}\{X \notin \mathcal{K}_m\}] - \mathbb{E}_{P_n} [f^{c_a}(X)] \\
\geq \; & \mathbb{E}_{P_{\mathcal{K}_m}} [f(X)] - \mathbb{E}_{P_n} [f^{c_a}(X)] - \epsilon(a+1) - \epsilon K^2(a+2) \\
\geq \; & \mathcal{D}_m - \epsilon \left( 1 + a + 1 + K^2(a+2) \right).
\end{aligned}
$$

By the arbitrariness of $\epsilon$, we complete the proof for the bounded cost function.

### Appendix A.4.3 Unbounded Cost Function

The following lemma is useful for finishing the last part of the proof.

**Lemma A4.** *Let $c_a(\cdot) = \min\{a, c(\cdot)\}$. For any $\epsilon$, let $P_{(a)}^\epsilon$ be an $\epsilon$-optimizer for the problem*

$$\inf_{P \in \mathcal{P}(\mathbb{R}^d)} \left\{ \mathcal{D}_{c_a}(P_n, P) : \mathbb{E}_{P_*}[f(X)] = \mathbb{E}_P[f(X)] \text{ for all } f \in \mathcal{B}(\mathbb{R}^d) \right\}.$$

*Then, the set $\left\{ P_{(a)}^\epsilon \right\}_{a=1}^\infty$ is relatively compact in the space $\mathcal{P}(\mathbb{R}^d)$ equipped with the topology of weak convergence.*

*Proof of Lemma A4.* First, we have

$$\mathcal{D}_{c_a}(P_n, P_{(a)}^\epsilon) \leq \mathcal{D}_{c_a}(P_n, P_*) + \epsilon \leq \mathcal{D}_c(P_n, P_*) + \epsilon < \infty.$$

If the set $\left\{ P_{(a)}^\epsilon \right\}_{a=1}^\infty$ is not relatively compact, then by Prokhorov's Theorem, there exists $\epsilon' > 0$ such that, for any compact set $\mathcal{K}$ and any $a_0 > 0$, we can find an $a > a_0$ with $P_{(a)}^\epsilon(\mathcal{K}) > \epsilon'$.

We pick $a_0 = \lceil (\mathcal{D}_c(P_n, P_*) + \epsilon) / \epsilon' \rceil$ and a sufficient large $\mathcal{K}$ such that

$$\inf_{x \in \mathcal{K}^c, y \in \{X_i\}_{i=1}^n} c(x,y) > a_0.$$

Then, for any $a > a_0$, we have

$$\mathcal{D}_{c_a}(P_n, P_{(a)}^\epsilon) > a_0 \epsilon' \geq \mathcal{D}_c(P_n, P_*) + \epsilon,$$

which leads to a contradiction. $\qquad\qquad\qquad\qquad\qquad\qquad\qquad\qquad\qquad\qquad\quad\square$

Next, we define the space $\Pi\left(P_n, P_*, \mathcal{B}(\mathbb{R}^d)\right)$ as

$$\Pi\left(P_n, P_*, \mathcal{B}(\mathbb{R}^d)\right) := \left\{ \pi \in \mathcal{P}(\mathbb{R}^d \times \mathbb{R}^d) : \pi(A \times \Omega) = P_n(A), \pi(\Omega \times A) = P(A) \right.$$
$$\left. \text{for every measurable set } A \subset \mathbb{R}^d, \text{ and } \mathbb{E}_P[f(X)] = \mathbb{E}_{P_*}[f(X)] \text{ for all } f \in \mathcal{B}(\mathbb{R}^d) \right\}.$$

We then have

$$R_n = \inf_{\pi \in \Pi(P_n, P_*, \mathcal{B}(\mathbb{R}^d))} \int_{\mathbb{R}^d \times \mathbb{R}^d} c(x,y) \pi(\mathrm{d}x, \mathrm{d}y).$$

Now let $I, I_a$ be respectively defined on $\Pi\left(P_n, P_*, \mathcal{B}(\mathbb{R}^d)\right)$ by

$$I_a[\pi] = \int_{\mathbb{R}^d \times \mathbb{R}^d} c_a(x,y) \pi(\mathrm{d}x, \mathrm{d}y) \qquad \text{and} \qquad I[\pi] = \int_{\mathbb{R}^d \times \mathbb{R}^d} c(x,y) \pi(\mathrm{d}x, \mathrm{d}y).$$

By Appendix A.4.2, we obtain

$$\inf_{\pi \in \Pi(P_n, P_*, \mathcal{B}(\mathbb{R}^d))} I_a[\pi] = \sup_{f \in \mathcal{LB}(\mathbb{R}^d)} \left\{ \mathbb{E}_{P_*}[f(X)] - \mathbb{E}_{P_n}[f^{c_a}(X)] \right\}.$$

We conclude the argument by showing that

$$\inf_{\pi \in \Pi(P_n, P_*, \mathcal{B}(\mathbb{R}^d))} I[\pi] = \sup_a \inf_{\pi \in \Pi(P_n, P_*, \mathcal{B}(\mathbb{R}^d))} I_a[\pi] \qquad\qquad (A.6)$$

and that, for each $a$,

$$\sup_{f \in \mathcal{LB}(\mathbb{R}^d)} \left\{ \mathbb{E}_{P_*}[f(X)] - \mathbb{E}_{P_n}[f^{c_a}(X)] \right\} \leq \sup_{f \in \mathcal{LB}(\mathbb{R}^d)} \left\{ \mathbb{E}_{P_*}[f(X)] - \mathbb{E}_{P_n}[f^c(X)] \right\}. \qquad (A.7)$$

Then, by the combination of (A.6), (A.7) and the weak duality, we will have the desired results.

Since $\inf I_a$ is a nondecreasing sequence, bounded above by $\inf I$, we only need to prove that

$$\lim_{a \to \infty} \inf_{\pi \in \Pi(P_n, P_*, \mathcal{B}(\mathbb{R}^d))} I_a(\pi) \geq \inf_{\pi \in \Pi(P_n, P_*, \mathcal{B}(\mathbb{R}^d))} I(\pi).$$

Let $\pi_a^\epsilon$ be an optimal coupling between $P_n$ and $P_{(a)}^\epsilon$ defined in Lemma A4. By the tightness of $\left\{ P_{(a)}^\epsilon \right\}_{a=1}^\infty$, we have that the sequence $\{\pi_a^\epsilon\}_{a=1}^\infty$ is also tight. Therefore, by Prokhorov's Theorem,

we are able to extract a subsequence $\left\{\pi_{a_k}^\epsilon\right\}_{k=1}^\infty$, where $\pi_{a_k}^\epsilon$ converges weakly to a probability measure $\pi_*^\epsilon \in \mathcal{P}(\mathbb{R}^d \times \mathbb{R}^d)$ as $k \to \infty$, in the sense that for any bounded continuous function $\theta$ on $\mathbb{R}^d \times \mathbb{R}^d$

$$\int \theta(x,y) d\pi_{a_k}^\epsilon(\mathrm{d}x, \mathrm{d}y) \to \int \theta(x,y) d\pi_*^\epsilon(\mathrm{d}x, \mathrm{d}y).$$

From this we observe that $\pi_*^\epsilon \in \Pi\left(P_n, P_*, \mathcal{B}(\mathbb{R}^d)\right)$. Then, whenever $a \geq b$, one has

$$I_a\left[\pi_a^\epsilon\right] \geq I_b\left[\pi_a^\epsilon\right].$$

By the boundedness of $c_b(\cdot, \cdot)$, we obtain

$$\limsup_{a\to\infty} I_a\left[\pi_a^\epsilon\right] \geq \limsup_{a\to\infty} I_b\left[\pi_a^\epsilon\right] \geq I_b\left[\pi_*^\epsilon\right].$$

By monotone convergence, $I_b\left[\pi_*^\epsilon\right] \to I\left[\pi_*^\epsilon\right]$ as $b \to \infty$, and thus

$$\lim_{a\to\infty} \inf_{\pi \in \Pi(P_n, P_*, \mathcal{B}(\mathbb{R}^d))} I_a(\pi) \geq \limsup_{a\to\infty} I_a\left[\pi_a^\epsilon\right] - \epsilon \geq I\left[\pi_*^\epsilon\right] - \epsilon \geq \inf_{\pi \in \Pi(P_n, P_*, \mathcal{B}(\mathbb{R}^d))} I(\pi) - \epsilon.$$

Then, by the arbitrariness of $\epsilon$, we have the desired results and conclude the proof.

### Appendix A.5   Proof of Theorem 4

Define $\mathcal{D}_A(P, Q) = \mathcal{D}_c(P, Q)$ with cost function $c(x, y) = (x - y)^\top A(x - y)$ for any positive definite matrix $A$. Then, we have

$$R_n \leq \inf_{P \in \mathcal{P}(\mathbb{R}^d)} \left\{ \mathcal{D}_I(P, P_n) : \mathbb{E}_P\left[f(\theta_i^\top X)\right] = \mathbb{E}_{P_*}\left[f(\theta_i^\top X)\right], \forall f \in \mathcal{C}_b(\mathbb{R}), \theta_1, \ldots, \theta_K \in \mathbb{R}^d \right\},$$

where $K \leq d$ and $\theta_1, \ldots, \theta_K$ are linearly independent. We first find orthonormal vectors $\theta_{K+1}, \theta_{K+2}, \ldots, \theta_d$ such that $\theta_1, \theta_2, \ldots, \theta_d$ are linearly independent and thus are a basis of $\mathbb{R}^d$. Let $Y_i = \theta_i^\top X$ for $i = 1, 2, \ldots, d$, let $P_*^Y$ denote the distribution of $Y$, and let $P_n^Y$ denote the corresponding empirical distribution. Further let $C = [\theta_1, \theta_2, \ldots, \theta_d]^\top$, and then $Y = CX$. Therefore, we obtain

$$\begin{aligned}
R_n &\leq \inf_{P \in \mathcal{P}(\mathbb{R}^d)} \left\{ \mathcal{D}_I(P, P_n) : \mathbb{E}_P\left[f(\theta_i^\top X)\right] = \mathbb{E}_{P_*}\left[f(\theta_i^\top X)\right] \ \forall f \in \mathcal{C}_b(\mathbb{R}), i = 1, 2, \ldots, K \right\} \\
&= \inf_{P \in \mathcal{P}(\mathbb{R}^d)} \left\{ \mathcal{D}_A\left(P^Y, P_n^Y\right) : \mathbb{E}_{P^Y}\left[f(Y_i)\right] = \mathbb{E}_{P_*^Y}\left[f(Y_i)\right] \ \forall f \in \mathcal{C}_b(\mathbb{R}), i = 1, 2, \ldots, K \right\}
\end{aligned}$$

where $A = \left(CC^\top\right)^{-1}$. Then, notice that

$$\begin{aligned}
\mathcal{D}_A\left(P^Y, P_n^Y\right) &= \inf_{\pi \in \mathcal{P}(\mathbb{R}^d \times \mathbb{R}^d)} \left\{ \left( \int (y - v)^\top A(y - v)\pi(\mathrm{d}y, \mathrm{d}v) \right) \right. \\
&\qquad \left. : \int_{v \in \mathbb{R}^d} \pi(\mathrm{d}y, \mathrm{d}v) = P^Y(\mathrm{d}y), \int_{y \in \mathbb{R}^d} \pi(\mathrm{d}y, \mathrm{d}v) = P_n^Y(\mathrm{d}v) \right\}.
\end{aligned}$$

Let $\rho(A)$ denote the spectral radius of matrix $A$. We then have $\mathcal{D}_A\left(P^Y, P_n^Y\right) \leq \rho(A)\mathcal{D}_I\left(P^Y, P_n^Y\right)$ and

$$\begin{aligned}
R_n &\leq \rho(A) \inf_{P \in \mathcal{P}(\mathbb{R}^d)} \left\{ \mathcal{D}_I\left(P^Y, P_n^Y\right) : \mathbb{E}_{P^Y}\left[f(Y_i)\right] = \mathbb{E}_{P_*^Y}\left[f(X)\right] \ \forall f \in \mathcal{C}_b(\mathbb{R}), i = 1, \ldots, K \right\} \\
&= \rho(A) \sum_{i=1}^K \mathcal{W}_2^2\left(P_*^{Y_i}, P_n^{Y_i}\right), \tag{A.8}
\end{aligned}$$

where $P_*^{Y_i}$ and $P_n^{Y_i}$ are the push-forward measures of $P_*^Y$ and $P_n^Y$ from $\mathcal{P}(\mathbb{R}^d)$ to $\mathcal{P}(\mathbb{R})$ such that for any Borel set $A$ in $\mathbb{R}$

$$\begin{aligned}
P_*^{Y_i}(A) &= P_*^Y\left(\{x \in \mathbb{R}^d : x_i \in A\}\right), \\
P_n^{Y_i}(A) &= P_n^Y\left(\{x \in \mathbb{R}^d : x_i \in A\}\right).
\end{aligned}$$

Notice that $\rho(A) = \rho\left(\left(C_K C_K^\top\right)^{-1}\right)$ for $C_K = [\theta_1, \theta_2, \ldots, \theta_K]^\top$. Therefore, $\rho(A)$ does not depend on our choices of $\theta_{K+1}, \theta_{K+2}, \ldots, \theta_d$. Finally, by Theorem 1 in [8], we obtain for the Wasserstein distance in the one-dimension case:

$$\mathbb{E}\left[\mathcal{W}_2^2\left(P_*^{Y_i}, P_n^{Y_i}\right)\right] \leq C\left(\mathbb{E}\left[|Y_i|^{4+\epsilon}\right]\right)^{2/(4+\epsilon)}/\sqrt{n} \leq CM\left(P_*\right)^2 n^{-1/2}. \tag{A.9}$$

By substituting (A.9) into (A.8), we have the desired results.

# Appendix B  Appendix: Experiments

## Appendix B.1  Marr Wavelet Basis for the $f_j$

We call $\phi(t)$ as the "mother" function for the wavelet basis. A function $f(t)$ is said to be written using a continuous wavelet basis as

$$f(t) = \int_s \int_u \frac{w_{s,u}}{\sqrt{s}} \phi\left(\frac{t-u}{s}\right) \mathrm{d}u \mathrm{d}s \quad \text{with } w_{u,s} = \int_t f(t) \frac{1}{\sqrt{s}} \phi\left(\frac{t-u}{s}\right) \mathrm{d}t,$$

where the index $u$ is called the translation, index $s$ the scaling, and $w_{u,s}$ the weights. Let $b_{u,s}(t) = \phi((t-u)/s)/\sqrt{s}$. Note that if $f(t)$ is a density then $w(u,s) = \mathbb{E}[b_{u,s}(t)]$. We will use a discrete version of this as our truncated basis, with a discrete set of $\{l = (u,s)\}_{l=1}^{L}$. Each $f_j$ is thus represented as

$$f_j(v) = \sum_{l=1}^{L} w_{jl} b_l(v),$$

where the truncated sequence of the $J$ terms gets us a finite dimensional variable $w = (w_{jl}, \ j = 1, \ldots, K, \ l = 1, \ldots, L)$.

The discrete basis set is determined by user input on a desired domain $[-M, M]$ for the function approximation and a *granularity* value $G$. If $[-m_0, m_0]$ is the domain of the mother function $\phi$, then the discrete pairs are determined as:

$$(u,s) = (k m_0 2^{-g+1}, 2^g), \quad \forall\, k \in \{-K(g), \ldots, K(g)\}, \ g = 0, \ldots, G, \qquad \text{(B.10)}$$

where $K(g) = \left\lceil \frac{M}{m_0 2^{-g}} \right\rceil + 1$. These finite set of pairs of $(u,s)$ over all $g, k$ are used to constitute the index $l \in \{1, \ldots, L\}$.

We need a basis that yields relatively smooth values for the derivative:

$$\frac{\mathrm{d}f_j}{\mathrm{d}v}(v) = \sum_{l=1}^{L} w_{jl} \frac{\mathrm{d}b_l}{\mathrm{d}v}(v), \qquad \text{(B.11)}$$

and thus we do not use the popular Haar basis, which yields $db_j/dv = 0$ a.e. We experiment with the Marr basis, also termed the (inverted) Mexican hat basis:

$$\phi(t) = \frac{2}{\sqrt{3}\pi^{1/4}}(1 - t^2)e^{-t^2/2}, \quad \text{with} \quad \frac{\mathrm{d}\phi(t)}{\mathrm{d}t} = \frac{2}{\sqrt{3}\pi^{1/4}}(t^3 - 3t)e^{-t^2/2}. \qquad \text{(B.12)}$$

## Appendix B.2  Re-formulation of $f^c(x)$

For each $x$, we have that

$$f^c(x) = \sup_{\Delta} \left( \sum_{j=i}^{K} f_j(\theta_j^\top(x + \Delta)) - \|\Delta\|_2^2 \right).$$

By Substituting $z = C_K \Delta$, for $C_K$ defined in Appendix A.5, we obtain

$$f^c(X_0) = \sup_{\Delta \in \mathbb{R}^d, v \in \mathbb{R}^K} \left( \sum_{j=i}^{K} f_j(\theta_j^\top x + v^{(j)})) - \|\Delta\|_2^2, \ s.t. \ v = C_K \Delta \right).$$

Keeping $v$ fixed and maximizing only over $\Delta$, we can see that the $\Delta^*(z)$ is the projection of the origin onto the linear subspace $C_K \Delta = z$. We can formally establish this by considering the Lagrangian formulation:

$$L(v, \Delta, \lambda) = \sum_{j=1}^{K} f_j(\theta_j^\top x + z^{(j)}) - \|\Delta\|_2^2 + \lambda^\top(C_K \Delta - z).$$

Setting up the first order optimality conditions, we have:

$$
\begin{aligned}
\nabla_z L = \nabla_z f_j(C_K x + z) - \lambda && = 0, \\
\nabla_\Delta L = -2\Delta + C_K^\top \lambda && = 0, \\
\nabla_\lambda L = C_K \Delta - z && = 0,
\end{aligned}
$$

where $\nabla_z f(C_K x + z) = [f_1'(\theta_1^\top x + z^{(1)}), \ldots, f_K'(\theta_K^\top x + z^{(K)})]^\top$. Taking the last two equations, and recalling that $\Gamma_K = C_K C_K^\top$, we obtain that

$$
\begin{aligned}
\Delta &= \frac{1}{2} C_K^\top \lambda, & C_K \Delta &= \frac{1}{2} \Gamma_K \lambda = z, \\
\lambda &= 2(\Gamma_K)^{-1} z, & \Delta &= C_K^\top (\Gamma_K)^{-1} z, \\
\|\Delta\|_2^2 &= z^\top (\Gamma_K)^{-1} z.
\end{aligned}
$$

Substituting in the first equation renders the first order equation to be satisfied as follows:

$$
\nabla_z f(C_K x + z) = 2(\Gamma_K)^{-1} z, \tag{B.13}
$$

which is also the first order condition for maximizing

$$
\sup_z \sum_{j=1}^{K} f_j(\theta_j^\top x + z^{(j)}) - z^\top (\Gamma_K)^{-1} z. \tag{B.14}
$$

This is an optimization problem in $z$, a $K$-dimensional variable. Hence, the true complexity of the inner supremum is a $K$-dimensional problem. The rows of $C_K$ are linearly independent by selection, so the inverse $(\Gamma_K)^{-1}$ exists.

### Appendix B.3  Implementation of $\hat{R}_n$ Computation

The $\hat{R}_n$ problem has a suggestive sup-inf form: the $w_{jl}$ variables attempt to emphasize the mass under $P_*$ to maximize the expectation $\mathbb{E}_{P_*}$, while the $z_n$ variables provide a mechanism for the $P_n$ samples to attain the same high values by also moving themselves, thus negating the value of the first term but at a quadratic penalty cost. This suggests why $R_n \to 0$ as $n$ grows and $P_n$ is sampled from $P_*$: the variables $z_n$ are able to attain the same expectations under $P_n$ with low cost. On the other hand, if $P_n$ comes from a different distribution than $P^*$, the weights $w_{jl}$ have more leeway to emphasize the non-overlapping parts of $P_*$, thus driving the supremum higher.

The inner supremum over $z$ is solved, as mentioned in the main body of the paper, using Newton iterations. Following (B.13), the Hessian can similarly be obtained as

$$
\nabla_z^2 f(C_K x + z) - 2(\Gamma_K)^{-1}.
$$

The problem (B.14) is a general non-linear optimization problem, and thus the Newton iterations return only locally optimal solutions. In order to obtain globally optimal solutions, this algorithm is restarted multiple times with randomized starting points. A particularly good set of initial starting points corresponds to samples from the $P_*$ (in $\mathbb{R}^d$ space) since the purpose of the $v$ (and equivalently $\Delta$) is to successfuly improve its optimal function value by moving close to the support of $P_*$.

The (stochastic estimate of the) gradient with respect to $w_{jl}$ for the stochastic approximation (SA) iterations of the outer optimization problem is obtained as:

$$
G(w) = \frac{1}{M} \sum_{m=1}^{M} \left( \sum_{j=1}^{K} \sum_{l=1}^{L} b_l(\theta_j^\top X_m) \right) - \frac{1}{n} \sum_{i=1}^{n} \left( \sum_{j=1}^{K} \sum_{l=1}^{L} b_l(\theta_j^\top X_i + z_i^*) \right). \tag{B.15}
$$

### Appendix B.4  Algorithm

Here is the complete algorithm to solve the optimization problem defining $\hat{R}_n$ once for a given $P_n$.

- Given: set $\{X_i, \ i = 1, \ldots, n\}$ that form $P_n$, and a sampler for $P_*$.
- Given: gain sequence $\gamma_r$.

- Initialize wavelet weights $w^{(0)} = (w^{(0)}_{jl})$ uniformly from $[-1, 1]$.
- For $r = 1, \ldots$     (SA method for outer maximization of weights $w_{jl}$)
    1. Sample $\{X_m, \ m = 1, \ldots, M\}$ from $P_*$ to estimate expectations under $P_*$.
    2. Assemble $m$-th summand of the first term in (B.15) as:
    $$\left( \nabla_w \left( \sum_{j=1}^{K} f_j(\theta_j^\top X_m) \right) \right)_{jl} = \sum_{j=1}^{K} b_l(\theta_j^\top X_m)$$
    3. For each $i = 1, \ldots, n$:
        (a) Estimate optimal $\Delta_i^*$ using deterministic Newton iterations and forming gradients from (B.11) and (B.12).
        (b) Return the $i$-th summand in the second term of (B.15) as
        $$\left( \nabla_w \left( \sum_{j=1}^{K} f_j(\theta_j^\top (X_i + \Delta_i^*)) \right) \right)_{jl} = \sum_{j=1}^{K} b_l \left( \theta_j^\top X_i + z_i^{(j)} \right),$$
        for $l = 1, \ldots, L$, and $j = 1, \ldots, K$.
    4. Assemble gradient for outer SA as given in (B.15) from the components in Steps (2) and (3b) above
    5. Set $w^{(r)} = w^{(r-1)} - \gamma_r G_r(w^{(r-1)})$

## Appendix B.5   Experimental Setup Details

The $P_*$ target distribution is set to be an equal weight mixture of four $d = 20$ dimensional unit-covariance Gaussians. The centers of the four Gaussians are $[-1, \ldots, -1]$, $[-1, 1, -1, \ldots, 1]$, $[1, -1, 1, \ldots, -1]$ and $[1, 1, 1, \ldots, 1]$. The $P_*^{\text{alt}}$ is also an equal mixture of four Gaussians with their centers obtained by applying an arbitrarily sampled rotation matrix to the centers of $P_*$.

We use the Marr wavelet basis, setting $m_0 = 4.5$ and $G = 3$, which yields $L = 28$ from (B.10), and in turn $84$ weight parameters $w_{jl}$.

The SA iterations for the outer optimization of $w_{jl}$ were conducted with mini-batches of size 50. A gain sequence of $\gamma_r = 100/(100 + r)$ was used. For each empirical set $P_n$ or $P_n^{\text{alt}}$ (sampled from $P_*$ or $P_*^{\text{alt}}$), we ran the SA algorithm 5 times to compute $\hat{R}_n$ or $\hat{R}_n^{\text{alt}}$, which are respectively defined by

$$
\begin{aligned}
\hat{R}_n \quad &:= \quad \sup_{w_{jl}} \left[ \mathbb{E}_{P_*} \left( \sum_{j=1}^{K} \sum_{l=1}^{L} w_{jl} b_l(\theta_j^\top X) \right) \right. \\
&\qquad \left. - \mathbb{E}_{P_n} \left( \sup_{z \in \mathbb{R}^K} \left( \sum_{j=i}^{K} \sum_{l=1}^{L} w_{jl} b_l(\theta_j^\top X + z^{(j)}) - z^\top (\Gamma_K)^{-1} z \right) \right) \right],
\end{aligned}
$$

$$
\begin{aligned}
\hat{R}_n^{\text{alt}} \quad &:= \quad \sup_{w_{jl}} \left[ \mathbb{E}_{P_*} \left( \sum_{j=1}^{K} \sum_{l=1}^{L} w_{jl} b_l(\theta_j^\top X) \right) \right. \\
&\qquad \left. - \mathbb{E}_{P_n^{\text{alt}}} \left( \sup_{z \in \mathbb{R}^K} \left( \sum_{j=i}^{K} \sum_{l=1}^{L} w_{jl} b_l(\theta_j^\top X + z^{(j)}) - z^\top (\Gamma_K)^{-1} z \right) \right) \right]
\end{aligned}
$$

and took their averages. This procedure seeks to average out the noise experienced in the SA method due to the fixed batch size of $50$ in estimating expectations under $P_*$.