[Reviews · NeurIPS 2020]

Review 1

Summary and Contributions: ***** UPDATE ***** I realize I might have been harsh in my evaluation. I believe the paper would have been more suited for a more theory oriented statistics conference / journal, but this is a recurrent problem in NeurIPS and I shouldn't have taken it out on the authors. While their theoretical result is really interesting, I also didn't appreciate that the authors barely mentioned previous work on statistical learning bounds with optimal transport. There have been recent efforts on the topic by several teams, and they should at least acknowledge them. However, if other reviewers took the time to thoroughly review the proof of the main result, I'm willing to increase my score. ****** The authors provide a through statistical study of the Wasserstein distance of an empirical measure to a given hypothesis class (i.e. a given set of functions). Thanks to a dual formulation of the problem, for which strong duality holds, they derive convergence rates for that distance. In particular, in the compact case they get a central limit theorem, which allows them to do hypothesis testing. They eventually test their method on a toy two-sample problem.

Strengths: The main strength of this paper is its thorough analysis of the distance it considers from a statistical standpoints. The distributional limit they obtain could be very useful for two sample testing with the Wasserstein distance.

Weaknesses: While the paper is really thorough on the theoretical aspects of the paper, I believe the main motivation of the paper (two sample testing) is not clearly explained. In section 4, where the authors use their theoretical results to perform two-sample tests, I did not understand how they proceed and how they choose to accept or reject the hypothesis.

Correctness: I haven't checked the theoretical claims in details. However, the methodology seems correct.

Clarity: In my opinion the paper lacks clarity. While the paper exhibits impressive theoretical results, it doesn't make a good job at explaining how they can be used by the community. Specifically, I found paragraph 3 in the introduction very vague, and should focus on a simple exemple to make their point. Mostly, section 4 should be rewritten so that the reader can understand how to apply the theoretical results for hypothesis testing (see questions below for specific issues).

Relation to Prior Work: I believe there should be a discussion on how this result relates to standard convergence results for the Wasserstein distance, which suffers from a curse of dimension. They should also mention other works which break the curse of dimension thanks to the cardinality of the space (Sommerfeld et al) or the regularization (Genevay et al).

Reproducibility: No

Additional Feedback: Questions: - In practice, do you use theorem 2 or theorem 4 for the two-sample problem? How do you choose which one you use? - How do you derive the rejection threshold from your theorems in section 4? - How do you compute the 95% quantile for Rn? I encourage the authors to rewrite the application / empirical section, to clarify how to apply their theoretical results for hypothesis testing.


Review 2

Summary and Contributions: The paper studies computing the Wasserstein distance between a probability measure and a set of of probability measures, where two distributions belong to the same set (are equivalent) if they have the same expectation over a set of specified transformations. The authors establish weak duality of such distance with a tractable expression involving a function transform, and strong duality under some constraints (compactness of the domain, and two other non-compact examples). There are many examples throughout the text particularizing the results to specific examples, and some simulations illustrating the convergence rate results.

Strengths: I find the contribution is strong as it addresses an important problem, which is the difficulty of approximating empirical distributions to continuous measures by limiting the classes of interest. The authors hypothesize and later show that limiting the projection to a set of probability measures improves the number of required samples to learn approximate distributions. This seems very relevant to ML applications and makes intuitive sense too. Furthermore, the authors connect their analysis with existing results, provide multiple illustrative examples, and some numerical validation results.

Weaknesses: I did not find clear weaknesses on this submission, the contribution seems reasonable and complete. The main weakness may be to further extend the practical impact of the analysis, but this is something that the authors address in the broader impact section, and something they are interested in pursuing. To the discretion of the authors, on a revised version of the manuscript I would suggest them to briefly expand the discussion on rates of convergence compared to normal convergences, but this is minor comment and the provided rates remain clear nonetheless.

Correctness: The claims and method seem correct, although I did not verify the proofs. The empirical methodology also seems correct and supports results and intuition.

Clarity: Yes, the paper is clear and well written.

Relation to Prior Work: Yes, relevant references are discussed, and comparisons with known results are present as well.

Reproducibility: Yes

Additional Feedback: A few minor typos: - page 4 line 152: wavet --> wavelet - page 7 line 245: othe --> of the - page 7 line 246: "(...) technique for constructing improving dual functions (...)" AFTER REBUTTAL: I thank the authors for their responses. I will maintain my original review of strong accept.


Review 3

Summary and Contributions: This paper considers estimating a constrained Wasserstein distance by constraining the test functions to a given class. Strong duality results are established for both compact and non-compact domains. Asymptotic properties of the constrained Wasserstein distance when the test function class involving linear projections are developed. Finally, preliminary numerical experiments are conducted for simulated mixtures of Gaussians. Edit after rebuttal: I am satisfied with authors' responses and will maintain my score of strong accept.

Strengths: (1) Both the strong duality and statistical convergence results are novel and generalize the Kantorovich duality in optimal transport and duality for Wasserstein distributionally robust optimization. The theoretical development are highly non-trivial and demonstrates the authors' master on techniques from infinite-dimensional optimization, statistical theory, and optimal transport theory. The derived results have an important impact and strengthen our understanding of both areas of distributionally robust optimization and optimal transport. (2) The authors pay close attention to the mathematical rigor of their derivations. (3) Numerical experiments on synthetic datasets confirm the theoretical findings.

Weaknesses: (1) The statistical convergence is established only for function classes involving a one-dimensional linear projection. The result is good but sort of expected, and hence makes the result not very strong. Can you establish results for general function classes in terms of its complexity? (2) The problem formulation (1) and its reformulation (3) remind me of the definition of the Integral Probability Metric (IPM), which is defined as the maximum difference of expectations of the test functions under the two compared distributions. It would be helpful to relate to and compare with IPM in terms of their respective pros and cons in terms of computational tractability, application scope, etc? (3) Can you conduct more experiments on real datasets? The current numerical experiments only consider simulated data from a mixture of Gaussians. It would be useful to see how the theoretical results are applied to real-world data such as image datasets with a more complicated structure.

Correctness: Yes.

Clarity: Yes. The paper is well organized and the results are clearly stated.

Relation to Prior Work: Yes.

Reproducibility: Yes

Additional Feedback:


Review 4

Summary and Contributions: This paper discovers the property of the Wasserstein distance as capturing the differece of distributions on a function class of interest. This duality result generalizes the Kantorovich-Rubinstein duality. This formulation also enjoys parametric convergence rate, beating the curse of dimensionality.

Strengths: As far as I checked, the theoretical claims in this paper are solid. I think the results in this paper are new and provides important insight on understanding how and why Wasserstein distance works.

Weaknesses: I didn't in particular find any major limitation of this work. Maybe few minor thing is that, I don't know if NeurIPS review process is suitable for checking this theoretically heavy paper (I didn't check all the proofs in time), and in appendix some math equations are off the margin.

Correctness: As far as I checked, the claims are correct. The experiments section also demonstrates the theoretical results.

Clarity: I think the paper is clearly written, although if there is an extra space I think it would be a good idea to make a conclusion section that discuss the results and concludes the work.

Relation to Prior Work: I think this work doesn't really have previous work to be directly compared, and related work is well discussed in Section 1.

Reproducibility: Yes

Additional Feedback: Minor typos: p.2, line 49: "an hypothesis" -> "a hypothesis" p.3, line 102: "an hypothesis" -> "a hypothesis" p.7, line 245: "the proof othe results" -> "the proof of the results" p.7, line 254: "[8, Theoerem 1]" -> "[8, Theorem 1]" p.8, line 278: "each iterations" -> "each iteration" p.8, line 281: "Guassians" -> "Gaussians"

[Author Response · NeurIPS 2020]

We thank all reviewers and we will modify the paper to clarify each of the points raised, as discussed/clarified below.

**R1**. (*Q3*) The primary contribution of our paper is a general theoretical framework for the analysis of Wasserstein
distance between an empirical measure and a probability measure when the hypothesis class is infinite dimensional. Our
goal is to provide insights to explain the empirical success of Wasserstein distance in practice, despite the well-known
curse of dimensionality. Two-sample testing is a specific application that can be addressed by our general framework,
but not the only one. For this specific two-sample testing case, we seek to verify the hypothesis test that an empirical
set $\{X_1, X_2, \ldots, X_n\}$ has been sampled from a given probability distribution $P_*$. We compute $\hat{R}_n$ and reject the
hypothesis if $\hat{R}_n$ is larger than the 95% quantile. The quantile is obtained by sampling $\hat{R}_n$ between $P_*$ and $P_n$, where
$P_n$ is the empirical distribution sampled from $P_*$. We will revise the presentation to make these points more clearly.

(*Q5*) The 4th paragraph of the introduction provides a concrete example of the high-level concepts discussed in the 3rd
paragraph, specifically describing how our theoretical results can be used in the discrimination step for constructing a
Wasserstein GAN. The questions concerning Section 4 and our theoretical results for hypothesis testing are addressed in
(*Q8*). We will refine the introduction and Section 4 to clarify how our theoretical results can be used by the community.

(*Q6*) While $R_n$ for infinite-dimensional function classes can scale as slowly as $O_p(n^{-1/d})$ (as discussed in the
paragraphs starting on L71 and L166), we establish that the rate of convergence for $R_n$ does not depend on the
dimension $d$ for the general class of infinite-dimensional functions considered in the paper, as formally presented in
Theorems 2 and 4 for the compact and non-compact settings, respectively. With respect to the literature you mentioned,
the work of Tameling et al. (2019) does recover parametric rates of convergence, but under the assumption that the
underlying measures are atomic. The entirety of Section 2.4 in Tameling et al. (2019) is dedicated to explaining (via the
development of a lower bound) why their method does not apply to continuous measures. The paper of Genevay et
al. (2017) is not a theoretical statistical analysis and it does not provide a rigorous rate of convergence for statistical
learning. We will include more discussion to clarify these points and comparisons with previous works.

(*Q8*) As discussed in our responses to (*Q3*) and (*Q5*), for the specific two-sample testing case, we seek to verify the
hypothesis test that an empirical set has been sampled from a given probability distribution. This includes, in both
the compact and non-compact settings, that we can exploit our strong duality results to compute an estimate for the
quantile of $R_n$ required to accept or reject a given hypothesis; refer also to (*Q3*) response. Theorem 2 provides a
characterization of the limiting distribution for $nR_n$ for the compact setting, which renders additional flexibility by
providing an alternate way to compute the required quantile. For the more complex non-compact setting, Theorem 4
establishes the convergence rate of $R_n$ to be $O_p(n^{-1/2})$. The primary goal of Theorem 4 is to establish a parametric
rate that provides insights into why Wasserstein distance can beat the curse of dimensionality in practice. Hence, we do
not use Theorem 4 to compute the specific quantile. We will refine the presentation to address all of these points more
clearly throughout the paper, including to clarify how to apply our theoretical results for hypothesis testing.

**R2 & R4**. Thank you both for your positive comments on our paper. As **R2** noted, we do plan to further extend the
practical impact of our theoretical analysis as part of ongoing research. To address the point raised by **R2**, we will
briefly expand the discussion on rates of convergence; and to address the points raised by **R4**, we will conclude with a
discussion of our theoretical results and planned future work, and we will refine the appendix to improve clarity and the
formatting of some of the equations. We also thank you both for the identified typos, which we will readily address.

**R3**. Thank you for your positive comments on our paper. (*1*) You raise a good point about our statistical convergence
results, which is well taken, with the caveat that we consider finitely many linear projections, not only a single one.
This, actually, makes the result very difficult to prove. To address your point, we will add a discussion about what hints
at the general condition to ensure the parametric rate, which, we believe, is closely related to the tightness of the formal
limiting object. This generalization is a topic of our current research. Note that Assumption 2 is a sufficient condition
given in terms of the problem primitives for this particular class. For more general function classes, we conjecture that
the convergence rate should be $O(n^{-1/d'})$, where $d'$ is the "effective dimension" (suitably defined) of the function class.
(*2*) You are indeed correct that IPM is similar to our formulation in terms of the duality representation. While there are
a few important technical differences, we note that it is not our primary intention to define a new metric. Rather we
seek to provide a thorough analysis of the Wasserstein distance, which has been the focus of a great deal of attention in
the statistical learning research literature. In particular, we add a new modeling feature, which is the hypothesis class
or the actor critic class. This induces a class of dual functions; and we note that our expression for the strong duality
(generalizing the celebrated Kantorovich-Rubinstein duality) uses the combination of both the function $f$ and its dual
$f^c$ in contrast with IPM. We thank you for raising this point, and we will include a discussion of this comparison in the
final version. (*3*) The primary contributions of our paper are theoretical in nature. However, as noted in our response to
**R2**, we plan to extend the practical impact of our theoretical work as part of ongoing research. This includes applying
our theoretical results to real-world dataset.

[Meta-Review · NeurIPS 2020]

Most of the reviewers were excited about this work, and I'm pleased to recommend it for publication. In the revision, please address all promised changes in the rebuttals and/or requested in the reviews. The outlier R1 has some valid points about the exposition as well as discomfort with the length of the appendix (it's true this is difficult to review in the NeurIPS environment), but these are not reasons to reject the work. That said, the authors of this paper are encouraged to take R1's expository suggestions seriously in their revision to make the work as approachable as possible.